# Mass Balance of the Greenland and Antarctic Ice Sheets from 1992 to 2020

Inès N. Otosaka[1], Andrew Shepherd[1], Erik R. Ivins[2], Nicole-Jeanne Schlegel[2], Charles Amory[3], Michiel R. van den Broeke[4], Martin Horwath[5], Ian Joughin[6], Michalea D. King[6], Gerhard Krinner[3], Sophie Nowicki[7], Antony J. Payne[8], Eric Rignot[9], Ted Scambos[10], Karen M. Simon[11], Benjamin E. Smith[6], Louise S. Sørensen[12], Isabella Velicogna[2,9], Pippa L. Whitehouse[13], Geruo A[9], Cécile Agosta[14], Andreas P. Ahlstrøm[15], Alejandro Blazquez[16], William Colgan[15], Marcus E. Engdhal[17], Xavier Fettweis[18], Rene Forsberg[12], Hubert Gallée[3], Alex Gardner[2], Lin Gilbert[19], Noel Gourmelen[20], Andreas Groh[5], Brian C. Gunter[21], Christopher Harig[22], Veit Helm[23], Shfaqat Abbas Khan[12], Christoph Kittel[3], Hannes Konrad[24], Peter L. Langen[25], Benoit S. Lecavalier[26], Chia-Chun Liang[9], Bryant D. Loomis[27], Malcolm McMillan[28], Daniele Melini[29], Sebastian H. Mernild[30], Ruth Mottram[31], Jeremie Mouginot[3], Johan Nilsson[2], Brice Noël[4], Mark E. Pattle[32], William R. Peltier[33], Nadege Pie[34], Ingo Sasgen[23], Himanshu V. Save[34], Ki-Weon Seo[35], Bernd Scheuchl[9], Ernst J.O. Schrama[36], Ludwig Schröder[5], Sebastian B. Simonsen[12], Thomas Slater[1], Giorgio Spada[37], Tyler C. Sutterley[38], Bramha Dutt Vishwakarma[39], Melchior van Wessem[4], David Wiese[2], Wouter van der Wal[11], Bert Wouters[11,4]

[1]Centre for Polar Observation and Modelling, University of Leeds, Leeds, United Kingdom
[2]Jet Propulsion Laboratory, California Institute of Technology, Pasadena, United States
[3]Institute of Environmental Geosciences, Université Grenoble Alpes, Grenoble, France
[4]Institute for Marine and Atmospheric Research, Utrecht University, Utrecht, The Netherlands
[5]Institut für Planetare Geodäsie, Technische Universität Dresden, Dresden, Germany
[6]Polar Science Center, University of Washington, Seattle, United States
[7]Department of Geology, University at Buffalo, Buffalo, United States
[8]School of Geographical Sciences, University of Bristol, Bristol, United Kingdom
[9]Earth System Science, University of California Irvine, Irvine, United States
[10]Earth Science and Observation Center, CIRES, University of Colorado Boulder, Boulder, United States
[11]Faculty of Civil Engineering and Geoscience, Delft University of Technology, Delft, The Netherlands
[12]National Space Institute, Technical University of Denmark, Lyngby, Denmark
[13]Department of Geography, Durham University, Durham, United Kingdom
[14]Laboratoire des Sciences du Climat et de l'Environnement, LSCE-IPSL, CEA-CNRS-UVSQ, Gif-sur-Yvette, France
[15]Glaciology and Climate, Geological Survey of Denmark and Greenland, Copenhagen, Denmark
[16]Spatial Geophysics and Oceanography Studies Laboratory, Toulouse, France
[17]ESA-ESRIN, Frascati, Italy
[18]Geography, University of Liège, Liège, Belgium
[19]Mullard Space Science Laboratory, University College London, West Sussex, United Kingdom
[20]University of Edinburgh, Edinburgh, United Kingdom
[21]Aerospace Engineering, Georgia Institute of Technology, Atlanta, United States
[22]Department of Geosciences, University of Arizona, Tucson, United States
[23]Glaciology, Alfred-Wegener-Institute Helmholtz-Center for Polar and Marine Research, Bremerhaven, Germany
[24]Satellite-based Climate Monitoring, Deutscher Wetterdienst, Offenbach/Main, Germany
[25]Department of Environmental Science, iClimate, Aarhus University, Roskilde, Denmark
[26]Department of Physics and Physical Oceanography, Memorial University, St. John's, Canada

[27]Geodesy and Geophysics Laboratory, NASA GSFC, Greenbelt, United States
[28]Lancaster Environment Centre, Lancaster University, Lancaster, United Kingom
[29]Istituto Nazionale di Geofisica e Vulcanologia, Rome, Italy
[30]SDU Climate Cluster, University of Southern Denmark, Odense, Denmark
[31]Research and Development Department, Danish Meteorological Institute, Copenhagen, Denmark
[32]isardSAT, Guildford, United Kingdom
[33]Physics, University of Toronto, Toronto, Canada
[34]Center for Space Research, University of Texas at Austin, Austin, United States
[35]Department of Earth Science Education, Seoul National University, Seoul, Korea
[36]Department SpE, Faculty of Aerospace Engineering, TU Delft, Delft, The Netherlands
[37]Dipartimento di Fisica e Astronomia, Alma Mater Studiorum Università di Bologna, Bologna, Italy
[38]Applied Physics Laboratory, University of Washington, Seattle, United States
[39]Interdisciplinary Centre for Water Research, Indian Institute of Science, Bengaluru, India

*Correspondence to*: Inès N. Otosaka (i.n.otosaka@leeds.ac.uk)

**Abstract.** Ice losses from the Greenland and Antarctic Ice Sheets have accelerated since the 1990s, accounting for a significant increase in global mean sea level. Here, we present a new 29-year record of ice sheet mass balance from 1992 to 2020 from the Ice Sheet Mass Balance Inter-comparison Exercise (IMBIE). We compare and combine 50 independent estimates of ice sheet mass balance derived from satellite observations of temporal changes in ice sheet flow, in ice sheet volume and in Earth's gravity field. Between 1992 and 2020, the ice sheets contributed $21.0 \pm 1.9$ mm to global mean sea-level, with the rate of mass loss rising from 105 Gt yr$^{-1}$ between 1992 and 1996 to 372 Gt yr$^{-1}$ between 2016 and 2020. In Greenland, the rate of mass loss is $169 \pm 9$ Gt yr$^{-1}$ between 1992 and 2020 but there are large inter-annual variations in mass balance with mass loss ranging from 86 Gt yr$^{-1}$ in 2017 to 444 Gt yr$^{-1}$ in 2019 due to large variability in surface mass balance. In Antarctica, ice losses continue to be dominated by mass loss from West Antarctica ($82 \pm 9$ Gt yr$^{-1}$) and to a lesser extent from the Antarctic Peninsula ($13 \pm 5$ Gt yr$^{-1}$). East Antarctica remains close to a state of balance with a small gain of $3 \pm 15$ Gt yr$^{-1}$, but is the most uncertain component of Antarctica's mass balance. The dataset is publicly available at https://doi.org/10.5285/77B64C55-7166-4A06-9DEF-2E400398E452 (The IMBIE Team, 2021).

# 1 Introduction

The Antarctic and Greenland Ice Sheets store the vast majority (99%) of Earth's freshwater ice on land. The rate of change in ice sheet mass - or ice sheet mass balance - is the net difference between mass loss through solid ice discharge at the grounding line, melting at the bed and at the ice-ocean interface and the surface mass balance (SMB; precipitation minus meltwater runoff, sublimation, evaporation, and erosion). Over the past three decades (between the 1990s and 2010s), ice losses from Antarctica and Greenland increased six-fold (The IMBIE Team, 2018, 2020), raising the global sea level (WCRP Global Sea Level Budget Group, 2018) and with it the risk of coastal flooding worldwide (Kulp and Strauss, 2019; Vitousek et al., 2017; Hanson et al., 2011). In Antarctica, the losses have arisen primarily due to ocean-driven melting of ice shelves (Adusumilli et

al., 2020; Paolo et al., 2015) and their collapse (Cook and Vaughan, 2010), which have accelerated the ice flow (Hogg et al.,
2017; Selley et al., 2021; Rignot et al., 2004), retreat (Konrad et al., 2018; Milillo et al., 2022; Jenkins et al., 2018) and
drawdown (Konrad et al., 2017; Shepherd et al., 2019) of numerous marine-terminating ice streams. In Greenland, increasing
air temperatures (Hanna et al., 2021) and decreasing cloud cover (Hofer et al., 2017) have exacerbated summertime surface
melting (Leeson et al., 2015; Tedesco and Fettweis, 2020) and runoff (Trusel et al., 2018; Slater et al., 2021), in tandem with
the speedup (Rignot and Kanagaratnam, 2006) and retreat (King et al., 2020) of outlet glaciers responding to a warming ocean
(Straneo and Heimbach, 2013). While ice sheet response to climate forcing remains the least constrained component of the
twenty-first-century sea level budget (Pattyn and Morlighem, 2020; Fox-Kemper et al., 2021), maintaining the long-term
observational record of ice sheet mass balance is critical to improving ice sheet model skill (Edwards et al., 2021; Ritz et al.,
2015) and confidence in projections of sea level rise (Aschwanden et al., 2021; Slater et al., 2020; Shepherd and Nowicki,
2017).

Thanks to the launch of new satellite missions and the development of improved geophysical corrections and models of SMB
and glacial isostatic adjustment (GIA), it is now possible to routinely monitor ice sheet mass changes using observations of
ice-flow derived from satellite radar and optical imagery (e.g. Gardner et al., 2018; Moon et al., 2012; Mouginot et al., 2017),
surface elevation changes (derived from satellite altimetry) (e.g. Sandberg Sørensen et al., 2018; Smith et al., 2020), and
fluctuations in Earth's gravity field (derived from satellite gravimetry from GRACE and its follow on) (e.g. Tapley et al., 2019;
Velicogna et al., 2020; Sasgen et al., 2020). The Ice Sheet Mass Balance Inter-comparison Exercise (IMBIE) has shown that
there is good agreement between these satellite methods (Shepherd et al., 2012) and that combining independent satellite-based
ice sheet mass balance estimates reduces uncertainty in estimates of Greenland and Antarctica's contribution to sea level rise.
By adopting a common framework to support the comparison and aggregation of ice sheet mass balance estimates generated
by different participants, it is possible to assess differences between techniques and the impact of using different geophysical
corrections, SMB models, or GIA models in ice sheet mass balance estimation to produce a reconciled time-series of ice sheet
mass changes. SMB models are required for estimating the net mass balance in the input-output method while GIA models are
necessary to correct ice sheet mass balance estimates derived from satellite gravimetry and to a lesser extent those derived
from satellite altimetry. The GIA is the result of solid Earth mass redistribution caused by changes in ice mass since the last
glaciation. Gravimetry fields record the combined effect of mass redistribution due to the GIA and recent changes in ice sheet
mass balance. The GIA contribution therefore needs to be modelled separately and removed from the gravimetry fields,
especially since it is of the same order of magnitude as the ice sheet mass balance signal (Caron and Ivins, 2020; Sutterley et
al., 2014a). Altimetry elevation change estimates also need to be corrected for the GIA. However, contrary to gravimetry
estimates, altimetry estimates are less sensitive to GIA as it manifests as an uplift (or subsidence) rate of the order of a few
millimetres per year, much smaller than the elevation changes recorded. The most recent IMBIE assessments for the Antarctic
Ice Sheet and the Greenland Ice Sheet covered the periods 1992 to 2017 and 1992 to 2018, respectively, and reported a
combined contribution of 17.8 ± 1.8 mm to global mean sea level (GMSL) between 1992 and 2017 (The IMBIE Team, 2018,

2020). Here, we extend these records to cover the same extended period (1$^{st}$ January 1992 to 31$^{st}$ December 2020) for both ice sheets. In the rest of the paper, all of time periods cited refer to the period extending from 1$^{st}$ January of the first year quoted to 31$^{st}$ December of the second year quoted.

In the years since our most recent assessment there have been notable changes in ice sheet mass in both hemispheres, and in the availability of satellite observations and ancillary datasets with which to detect these changes. In Greenland, for example, atmospheric blocking and reduced summertime snowfall (Tedesco and Fettweis, 2020) led to near-record levels of meltwater runoff in 2019 (Slater et al., 2021) which, in combination with progressively increasing ice discharge (Mouginot et al., 2019), set a new record for annual ice losses during the satellite era (Sasgen et al., 2020). In Antarctica, pervasive mass losses have

continued in the Amundsen Sea Sector (Groh and Horwath, 2021) as a consequence of further grounding line retreat (Milillo et al., 2022) and the associated glacier speedup (Joughin et al., 2021). A follow on to the GRACE satellite mission (GRACE-FO) was launched in May 2018 (Tapley et al., 2019), the ICESat-2 satellite laser altimeter mission was launched in September 2018 (Smith et al., 2020), and updated products have been released for many others - including swath altimetry from CryoSat-2 (Gourmelen et al., 2018). To accompany these observations, there have been updated models of GIA (e.g. Caron and Ivins,

2020) to correct mass and elevation changes associated with solid earth movement, of firn densification (e.g. Stevens et al., 2020) to correct changes in elevation for surface processes, and of SMB (e.g. Fettweis et al., 2020; Mottram et al., 2021) to aid mass budget and mass balance partitioning calculations.

Here, we make use of new satellite observations, new methods and models to provide an updated IMBIE assessment of Greenland and Antarctic ice sheet mass balance, extending our most recent records by 3 and 4 years, respectively. We provide

a description of the datasets incorporated in this updated assessment and of the aggregation methods employed. We also discuss differences between the ice sheet mass balance estimates derived from altimetry, gravimetry and the input-output method, and we present extended reconciled time-series of ice sheet mass change. We discuss the limitations of our dataset and outline a roadmap for future improvements. Finally, we contrast our findings with trends in GMSL and compare them with projections of future ice sheet mass changes from the Intergovernmental Panel on Climate Change's (IPCC) Sixth Assessment Report

(AR6).

## 2 Data

Fluctuations in ice sheet mass are a key indicator of ice sheet stability and can be inferred using a range of satellite techniques (Shepherd et al., 2012). Satellite altimetry measures ice sheet elevation change, computed at orbit crossing points by calculating the difference in ice sheet elevation at a crossover point between ascending and descending satellite passes (e.g. Wingham et

al., 1998), using clusters of data points acquired along all ground tracks (e.g Pritchard et al., 2009), or by differencing height models separated over time (e.g. Csatho et al., 2014). Mass balance is estimated by accounting for changes in bedrock elevation (e.g. Caron and Ivins, 2020) and then by either prescribing the density associated to the elevation fluctuation (e.g. Shepherd et

al., 2019) or by making a model-based correction for changes in firn compaction (Sørensen et al., 2011). The technique is unique in charting patterns of mass imbalance with fine (monthly) temporal sampling and fine ($10^2$ km$^2$) spatial resolution,

and there are continental-scale measurements dating back to the early 1990s. Satellite measurements of ice velocity computed from sequential radar and optical imagery (e.g. Rignot and Kanagaratnam, 2006) are the basis of ice sheet input-output assessments (e.g. Rignot et al., 2019; Mouginot et al. 2019). Ice velocities are combined with estimates of ice thickness (e.g. Morlighem et al., 2017) to compute changes in marine-terminating glacier discharge, and then with regional climate model estimates of surface mass balance sources (snowfall, rainfall) and sinks (runoff, sublimation, evaporation, and erosion) (e.g.

Fettweis et al., 2020; Mottram et al., 2021) to measure temporal changes in net mass balance. The technique provides monthly to annual temporal sampling and drainage basin scale spatial resolution, and there are continental-scale measurements dating back to the late 1970s. During the last decade, new satellite missions with a more frequent revisit time (down to 6 days using image pairs from Sentinel-1a and Sentinel-1b available during the period 2016 to 2021 until the end of Sentinel-1b mission) have been used to improve the temporal resolution of ice velocity measurements, allowing to investigate seasonal fluctuations

in ice velocity (King et al., 2018; Lemos et al., 2018) and produce monthly estimates of ice discharge at the continental scale. Mankoff et al. (2021) even produced daily estimates of ice sheet mass balance from the input-output method by resampling the velocity data, however the original temporal resolution of ice velocity measurements does not exceed 12 days. Satellite gravimetry measures fluctuations in Earth's gravitational field, computed using either global spherical harmonic solutions (e.g. Velicogna and Wahr, 2006) or using spatially discrete mass concentration units (e.g. Luthcke et al., 2006). Ice sheet mass

changes are determined after making model-based corrections for GIA (e.g. Caron and Ivins, 2020) and for the leakage of mass trends occurring elsewhere in the climate system, especially those arising from ocean mass variability and changes in land hydrology. The technique provides fine (monthly) temporal sampling and moderate ($10^5$ km$^2$) spatial resolution, dating back to 2002 with the launch of the GRACE mission and the more recent launch of its follow on GRACE-FO in 2018.

## 2.1. Input Data

To compile our assessment of Greenland Ice Sheet mass balance we use 27 satellite-based estimates of ice sheet mass change, including 8 estimates based on satellite altimetry, 16 based on satellite gravimetry, and 3 based on the input-output method. Compared to the most recent IMBIE assessment, 12 of these estimates have been updated to include more recent data for Greenland. This set of updated estimates is made of 2 estimates from the input-output method, 1 altimetry estimate, and 9 gravimetry estimates including data from the new GRACE Follow-On space gravimetry mission (GRACE-FO). For our

assessment of Antarctica's mass balance, we use 23 satellite-based estimates altogether, with 6 derived from altimetry, 16 from gravimetry, and 1 from the input-output method. More than half of these estimates have been extended in time compared to IMBIE-2. These updated estimates for Antarctica include the input-output method estimate, 2 altimetry estimates, and 10 gravimetry estimates combining GRACE and GRACE-FO data. In total, this new IMBIE assessment includes data from 14 satellite missions, spanning the years 1992 to 2020 – with results from all three geodetic techniques available between 2003

and 2018 in Greenland and 2002 and 2018 in Antarctica – and, for the first time, includes data from the GRACE-FO mission launched in 2018. A wide range of GIA models have been used to correct gravimetric and volumetric mass balance estimates. The models use in this assessment are all forward models, which combine a rheology model of the solid Earth with a model of past ice mass change. In this assessment, only two SMB models have been used in the input-output method estimates included – the RACMO (Regional Atmospheric Climate Model) and MAR (Modèle Atmosphérique Régional) models (Table

180 1).

> **Table 1. Synthesis of satellite datasets, GIA, and SMB models used to derive the individual estimates of ice sheet mass balance included in this study. Details and references of the GIA and SMB models are available in Appendix A.**

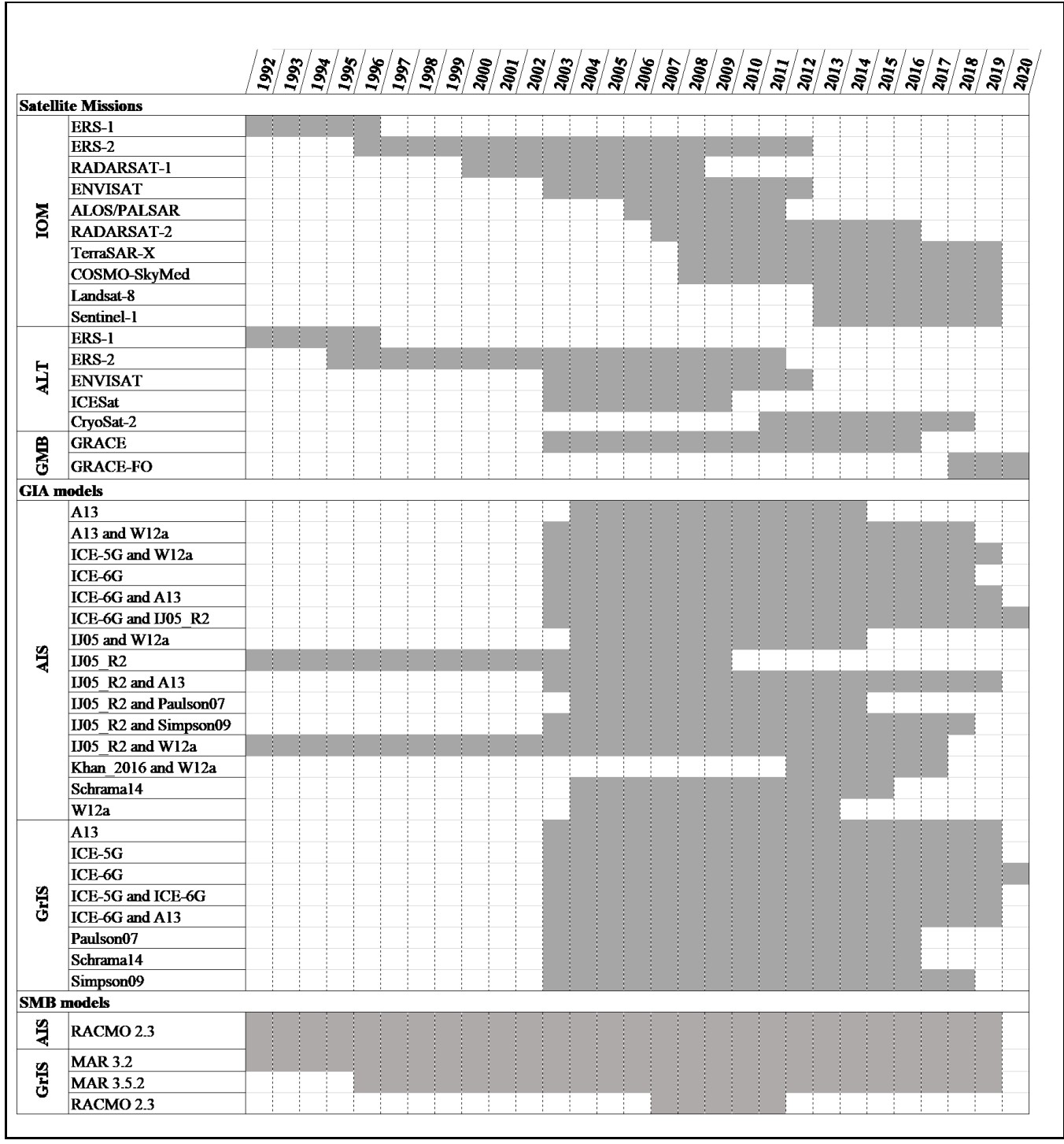

To achieve a meaningful comparison of ice sheet mass balance estimates, we analyse mass trends using common definitions of the Antarctic, West Antarctic, East Antarctic, Antarctic Peninsula, and Greenland Ice Sheet boundaries (AIS, WAIS, EAIS,

APIS, and GrIS, respectively). We use two ice sheet drainage basin sets, both previously used in the past IMBIE assessments (Shepherd et al., 2012; IMBIE Team, 2018; 2020). The first drainage basin set was derived based on ICESat surface elevation

data and includes 27 basins in Antarctica covering an area of 11,885,725 km$^2$ and 19 in Greenland over an area of 1,703,625 km$^2$ (Zwally et al., 2012) and is retained for consistency with the first IMBIE assessment (Shepherd et al., 2012). The second set defines 18 basins in Antarctica covering 11,892,700 km$^2$ and 6 in Greenland covering 1,723,300 km$^2$ (Rignot et al., 2011a; Rignot et al., 2011b). The two ice sheet delineation differ by 1.1 % and 0.1 % of total ice sheet extent for the Greenland and Antarctic Ice Sheets, respectively, and thus using either of these definitions leads to a negligible difference in mass balance

(The IMBIE Team, 2018; 2020). IMBIE participants were free to use either of these two definitions, and we combine mass trends over the GrIS, AIS, WAIS, EAIS, and APIS together regardless of what definition was chosen. The different estimates included in this assessment are presented on Figure 1.

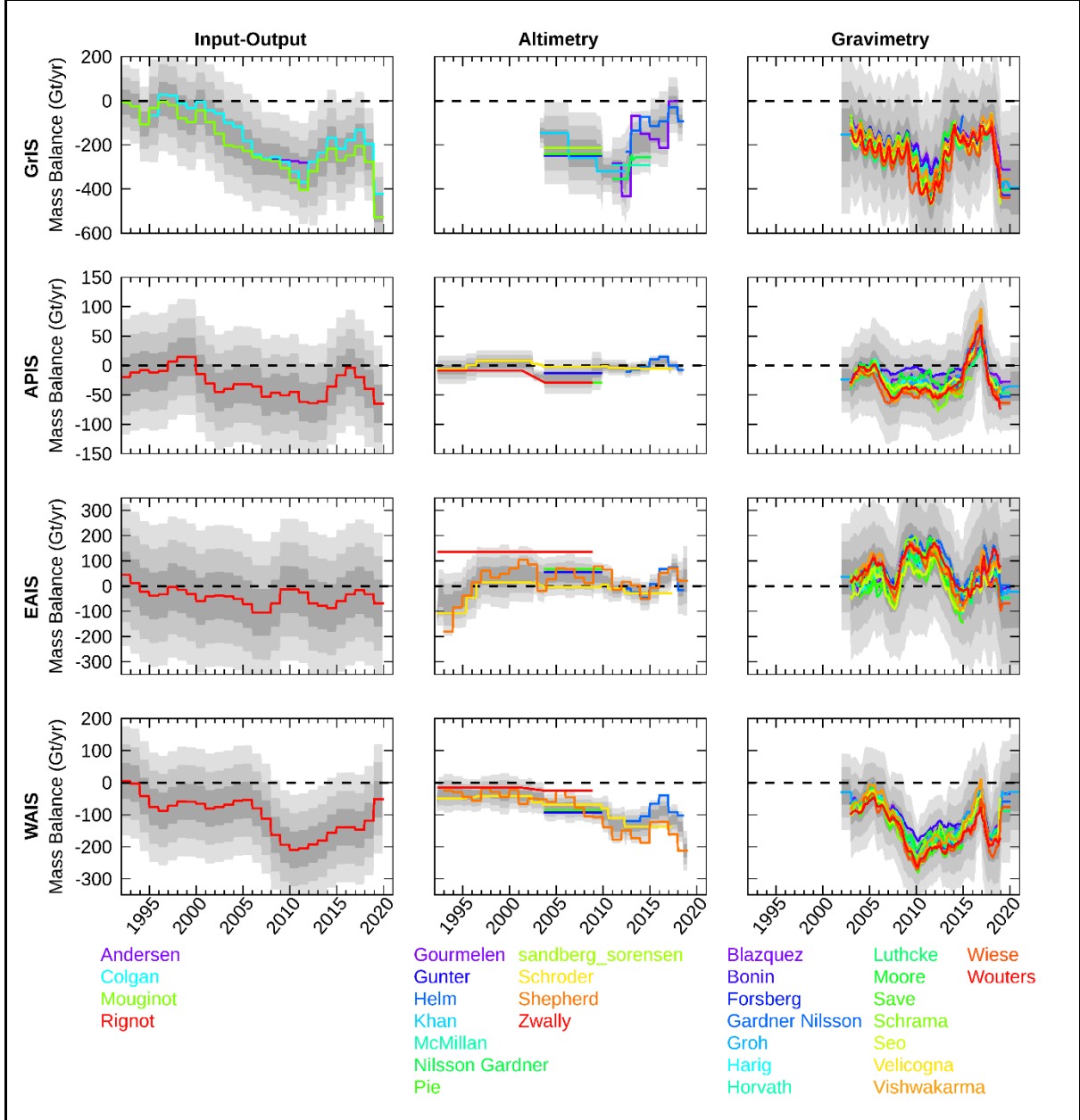

**Figure 1. Individual rates of ice sheet mass balance from the input-output, altimetry, and gravimetry groups over the GrIS, APIS, EAIS, and WAIS included in this study. The grey shading shows the estimated 1σ, 2σ, and 3σ ranges of the aggregated time-series per group in dark, mid, and light grey, respectively. The uncertainty is calculated as the root mean square of the contributing errors at each monthly epoch.**

**2.2 Output Data**

The output data consists of a single reconciled estimate of ice sheet mass balance covering the period 1$^{st}$ January 1992 to 31$^{st}$ December 2020 for the GrIS, AIS, APIS, WAIS, EAIS, and the sum of the GrIS and AIS. Two CSV files are provided for each ice sheet region, one with the data provided in Gigatons (Gt) and one with the data provided in equivalent sea level contribution in millimetres (mm). These files contain annual rates of mass balance and cumulative mass changes with their corresponding uncertainties.

**3 Methods**

IMBIE participants contributed time-series of either relative mass change, $\Delta M(t)$, or of rate of mass change, $dM(t)/dt$, with their associated uncertainty, integrated over at least one of the ice sheet regions defined in the standard drainage basin sets. To produce a reconciled estimate of ice sheet mass change from these individual estimates, we compare and aggregate $dM(t)/dt$ from each satellite technique. The IMBIE assessment software used to produce the dataset presented in this study is available

at https://doi.org/10.5281/zenodo.7342481. We apply a consistent processing scheme to all submitted datasets and for all ice sheet regions which consists of: i) computing $dM(t)/dt$ for all datasets that were submitted as $\Delta M(t)$, ii) aggregating time-series of mass trends within each class of satellite observations, iii) combining the altimetry, gravimetry, and mass budget time-series to derive a single reconciled time-series of mass trends, and iv) integrating this reconciled time-series of mass trends to produce the final reconciled time-series of cumulative mass change. In what follows, we summarise each of these processing steps:


*i) Computing time-series of mass trends*

First, we derive time-series of monthly rates of ice sheet mass change, $dM(t)/dt$, for all datasets that were submitted as $\Delta M(t)$ to allow the aggregation of datasets within each satellite observations class as $dM(t)/dt$ computed using a standardised approach. At each epoch, we estimate $dM(t)/dt$ by fitting a linear trend to the $\Delta M(t)$ data falling within a sliding window of 36

months, centred around the given epoch, using a weighted least-squares approach, with each point weighted by its error. The error on the derived time-series is taken as the regression error which incorporates the original measurement error and the linear model structural error computed as the standard error of the linear regression. Finally, the derived time-series of mass trends are truncated by half the window width at the start and end of their period.

*ii) Aggregating time-series of mass trends from similar satellite observations*

We aggregate the standardised time-series of mass trends within the altimetry, gravimetry, and mass budget groups separately to produce three time-series over each ice sheet region. We calculate each aggregated time-series by taking the error-weighted average of monthly rates of ice sheet mass change computed using the same technique. The associated error is calculated as the root mean square of the contributing time-series errors.

*iii) Combining the altimetry, gravimetry, and mass budget time-series of mass trends*

We combine the altimetry, gravimetry, and input-output time-series to produce a single reconciled time-series of mass trends by taking the error-weighted mean of the available estimates at each epoch. We estimate the error on the reconciled mass trend time-series at each epoch as the root mean square error divided by the square root of the number of independent techniques

for which a mass trend estimate is available. From this reconciled time-series of mass trends, we compute rates of mass balance over each calendar year and over different time periods as the average of the monthly rates falling within the defined time interval, with the associated error as the average of the contributing errors divided by the square root of the numbers of years of the time period. Finally, when summing mass trends of multiple ice sheets, the combined uncertainty is estimated as the root sum square of the uncertainties for each region.


*iv) Generating the final reconciled time-series of cumulative mass change*

We generate a time-series of cumulative ice sheet mass change by integrating our reconciled time-series of mass trends over time for each ice sheet. We estimate the cumulative errors as the root sum square of annual errors, assuming that errors are not correlated over time. Errors quoted in the text refer to the $1\sigma$ estimated error.

**4 Results**

First, we compare individual estimates of ice sheet mass balance within each of the three geodetic technique experiment groups, separately, to assess the level of agreement among estimates derived using the same technique. Within each group, we compare annual rates of mass change and their standard deviation for each ice sheet region. The input-output group includes significantly fewer mass balance estimates than the other technique experiment groups, but these estimates have the advantage of providing

information on the partitioning of mass trends between signals related to SMB and ice dynamics, and they also cover relatively long periods of time. Ice discharge is measured from satellite observations of ice velocities combined with estimates of ice thickness at glaciers' termini, and SMB is derived from regional climate model outputs. To estimate the SMB anomaly in Greenland, two estimates used MAR (version 3.2 and version 3.5.2) and one used RACMO (version 2.3). In Antarctica, the input-output estimate used RACMO (version 2.3). In addition to using different SMB models, those estimates also define

different reference periods to calculate the SMB anomalies. All of the mass balance estimates derived in this group were originally posted at annual resolution and we resample them over monthly epochs to aggregate them with estimates from the other groups. We include 3 input-output method estimates of GrIS mass balance, all at annual resolution and that together span the period 1992 to 2020 and overlap during the period 2007 to 2011. During their common period, annual rates of mass change determined from these three input-output datasets have a median difference of 28.5 Gt yr$^{-1}$ with a standard deviation of 35 Gt

yr$^{-1}$. For Antarctica and its ice sheet components, we include one input-output mass balance estimate which covers the entire 1992 to 2020 period at annual resolution.

The altimetry group includes 8 mass balance estimates for the GrIS that together span the years 2003 to 2018, with 4 of these solutions derived from radar altimetry, 2 from laser altimetry, and 2 from a combination of both. We include 6 altimetry mass balance estimates for the AIS which together cover the period 1992 to 2019. In total we include 6 solutions for the EAIS, 6 for the WAIS, and 5 for the APIS. Of these, 2 solutions are derived from radar altimetry, 1 from laser altimetry, and 3 from a combination of both. To derive rates of surface elevation change, various methods were applied to the laser and radar altimetry data including repeat-track, plane fit, or overlapping footprints techniques. For Greenland, half of the participants corrected the altimetry time-series for the GIA effect while for Antarctica, all participants applied a GIA correction. Next, to derive mass trends from rates of surface elevation change, either a constant density or a spatially and time varying density field from a firn density model forced by a regional climate model, were applied. These solutions have varying temporal resolutions ranging from 1 month to 7.1 yr for an average effective temporal resolution of 3.0 yr for Greenland and 2.6 yr for Antarctica. The temporal resolution of the altimetry group is thus lower than annual, mainly due to the fact that solutions derived from laser altimetry data were all provided as constant rates spanning the duration of ICESat-1 mission while the radar altimetry solutions have a higher temporal resolution of 0.35 yr for Greenland and 0.47 yr for Antarctica. As there is no overlap period during which all altimetry estimates are available, we compare solutions derived solely from radar altimetry and solutions incorporating laser altimetry data separately. In Greenland, radar altimetry solutions have a median difference of 144 Gt yr$^{-1}$ and standard deviation of 67 Gt yr$^{-1}$ during their two-year overlap period (2013 to 2014) while the median difference between laser and combination solutions is 29 Gt yr$^{-1}$ with a standard deviation of 29 Gt yr$^{-1}$ during their 6-year overlap (2004 to 2009). In Antarctica, the spread between laser solutions is largest at the EAIS with a standard deviation in annual rates of 38 Gt yr$^{-1}$ between 2004 and 2008, followed by the WAIS and APIS with standard deviations of 23 Gt yr$^{-1}$ and 10 Gt yr$^{-1}$, respectively. On the other hand, radar altimetry solutions show a larger spread at the WAIS (21 Gt yr$^{-1}$) than at the EAIS (14 Gt yr$^{-1}$) during their overlap period (2013 to 2018).

The gravimetry group has the largest number of estimates, with 16 for each ice sheet that together span the period 2002 to 2020. All gravimetry solutions were submitted as time-series of cumulative mass change at monthly resolution resulting in a collective effective resolution of 0.08 yr. All participants submitted estimates for all ice sheet regions, with 10 participants analysing spherical harmonic gravity field solutions using a wide range of approaches and 6 participants using mass concentration units (usually referred to as mascons) directly estimated from the GRACE and GRACE-FO level-1 K-band ranging data. Various GIA, hydrology leakage, and ocean leakage models were used to correct the gravimetry data for external signals. Overall, there is good agreement between rates of ice sheet mass balance derived from satellite gravimetry. In Greenland, we compare the different gravimetry solutions over the period 2012 to 2014 and find that annual rates of mass have a median difference of 36 Gt yr$^{-1}$, and standard deviation is 30 Gt yr$^{-1}$. In Antarctica, the different gravimetry solutions overlap over a decade from 2004 to 2014 during which their annual rates of mass balance have a median difference of 41 Gt yr$^{-1}$. When comparing over the different regions of the Antarctic continent, the difference is greatest at the EAIS with a median difference of 31 Gt yr$^{-1}$ and standard deviation of 26 Gt yr$^{-1}$. In the other regions, gravimetry estimates are in better agreement at the APIS

with a median difference of 8 Gt yr$^{-1}$ and standard deviation of 10 Gt yr$^{-1}$, followed by the WAIS where the median difference between estimates reaches 19 Gt yr$^{-1}$ and their standard deviation is 17 Gt yr$^{-1}$.

Comparing mass balance estimates derived from similar satellite observations reveals that in Greenland, the median difference between estimates is the largest for the altimetry group and the smallest for the input-output group. In Antarctica, the median difference between altimetry estimates is less than 38 Gt yr$^{-1}$ and less than 41 Gt yr$^{-1}$ for gravimetry estimates during their

respective overlap periods. However this comparison is limited by the varying temporal resolutions of the different datasets – especially for the altimetry group for which constant rates of mass change over long periods of time dampen temporal variation in ice sheet mass changes – and by the small number of input-output estimates – in particular in Antarctica where only one estimate is available. This limits our ability to link differences between estimates derived from the same geodetic technique to methodological differences, or to the use of different geophysical corrections or auxiliary datasets.

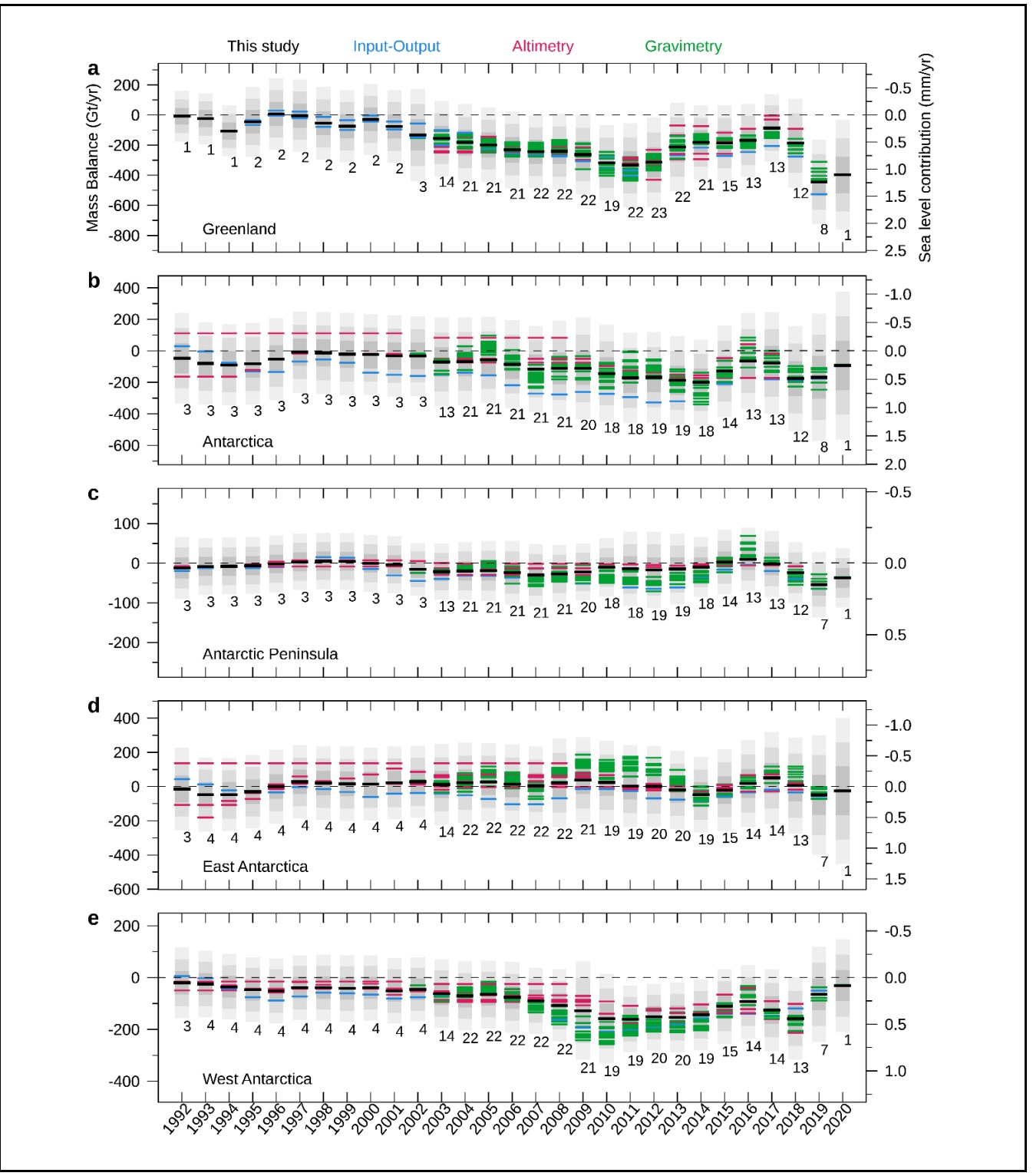

**Figure 2. Annual rates of mass change of the (a) GrIS, (b) AIS, (c) APIS, (d) EAIS, and (e) WAIS from the altimetry, gravimetry and input-output estimates included in this study (shown by the coloured bars) and the reconciled estimate produced from combining those estimates (shown by the thick black bars). The estimated 1σ, 2σ, and 3σ ranges of our final reconciled estimate are shaded in dark, mid and light grey, respectively. The number of individual mass balance estimates collated at each epoch is shown below each bar.**

Next, we assess differences between the aggregated time-series derived within each class of satellite observations during the periods when estimates from all three geodetic techniques are available – from 2003 to 2018 for Greenland and from 2002 to 2019 for Antarctica (Figure A1). We compare rates of mass change during these overlap periods, which are 5 and 10 years longer than in the previous IMBIE assessments, respectively (Figure 3). We report the standard deviation of the aggregated-altimetry, gravimetry and input-output estimates rates of mass change and compare it to the reconciled rate of mass change and its uncertainty (computed as described in Section 3). In Greenland, rates of mass balance determined from altimetry, gravimetry, and the input-output method are in close agreement between 2003 and 2018, with a standard deviation of 19 Gt yr$^{-1}$ and a reconciled rate of mass loss of 221 ± 22 Gt yr$^{-1}$ from all three techniques. In Antarctica, the reconciled rate of mass loss between 2003 and 2019 is 115 ± 24 Gt yr$^{-1}$ but the spread of the altimetry, gravimetry and mass budget estimates is 4 times larger than in Greenland (79 Gt yr$^{-1}$). Over the different regions of Antarctica, the spread of estimates of ice sheet mass balance increases with the size of the region considered, with standard deviations of 54 Gt yr$^{-1}$, 18 Gt yr$^{-1}$, and 16 Gt yr$^{-1}$, at the EAIS, WAIS, and APIS, respectively. Across all ice sheets, the input-output estimate is the most negative and the altimetry the most positive except at the EAIS, where the gravimetry estimate is the most positive. The greatest departure occurs at the EAIS where the three geodetic techniques disagree on even the sign of the mass change, with a maximum difference of 105 ± 33 Gt yr$^{-1}$ between rates of mass change from the input-output method and gravimetry estimates. This indicates that the EAIS remains a challenging region for which to monitor mass changes, likely due to the large extent of this region, the poorly constrained GIA signal and paleo-ice reconstruction (Bentley et al., 2014; Martín-Español et al., 2016; Small et al., 2019), and the relatively small mass imbalance in comparison to natural fluctuations in SMB in East Antarctica (Mottram et al., 2021).

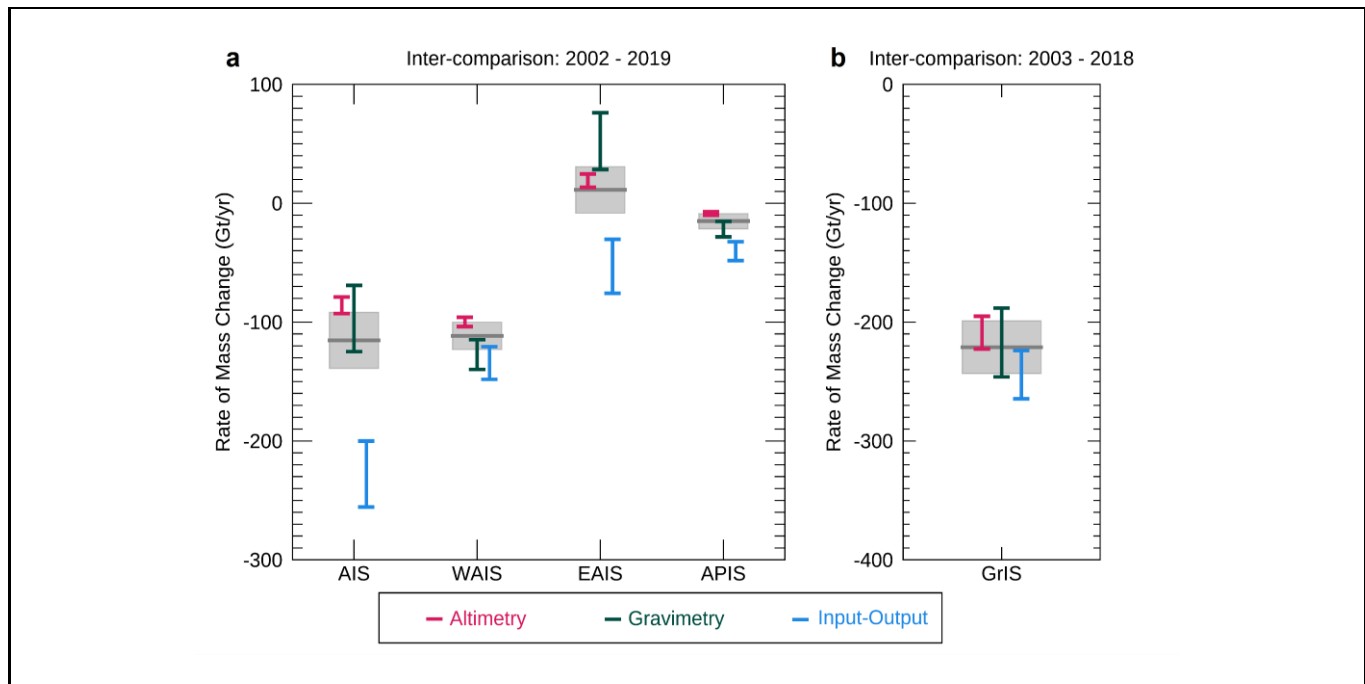

**Figure 3. Inter-comparison of rates of ice sheet mass balance of (a) the AIS, WAIS, EAIS, and APIS over the overlap period 2002-2019 and of (b) the GrIS during the overlap period 2003-2018 derived from the altimetry, gravimetry, and input-output techniques.**

When examining the aggregated time-series of rate of mass change at annual resolution, we find the highest temporal correlation between the three time-series at the WAIS ($0.6 < r^2 < 0.9$). In addition, the gravimetry and input-output annual rates are also well-correlated at the APIS and GrIS ($r^2 > 0.5$). However, the altimetry mass balance time-series is poorly correlated with both the aggregated gravimetry and input-output time-series at the APIS, EAIS, and GrIS ($r^2 < 0.2$). The better correlation between the gravimetry and input-output time-series can be explained by their higher temporal resolutions, sufficient to resolve annual fluctuations in ice sheet mass balance which are substantial in these regions. Nonetheless, we find that almost all individual estimates of annual rates of mass balance included in this study fall within one standard deviation ($1\sigma$) of our reconciled estimate given their respective individual errors, with 100 %, 96 %, 100 %, 96 %, and 99 % of those annual rates of mass change falling within $1\sigma$ at the GrIS, AIS, APIS, EAIS, and WAIS, respectively.

We integrate the combined mass balance estimates from gravimetry, altimetry, and the input-output method (Figure 2) to determine the cumulative mass lost from Antarctica and Greenland since 1992 (Figure 4). Antarctic mass loss continues to be dominated by ice discharge from West Antarctica where the signal is strongest – rising from $37 \pm 19$ Gt yr$^{-1}$ between 1992 and 1996 to a maximum of $131 \pm 21$ Gt yr$^{-1}$ between 2012 and 2016 (Table 2), before slowing slightly to $94 \pm 25$ Gt yr$^{-1}$ during the last 5 years of our survey between 2017 and 2020. At the Antarctic Peninsula the increase in losses since the early 2000s that

is generally associated with ice-shelf collapse (Rignot et al., 2004; Cook and Vaughan, 2010; Adusumilli et al., 2018) was
masked briefly between 2012 and 2016, when the average rate of mass loss was reduced by 15 Gt yr$^{-1}$ to $6 \pm 13$ Gt yr$^{-1}$ in part
due to an extreme snowfall event in 2016 (Wang et al., 2021; Chuter et al., 2021), before returning to $21 \pm 12$ Gt yr$^{-1}$ between
2017 and 2020. East Antarctica remains the least certain component of Antarctic Ice Sheet mass balance, where the average
30-year mass trend is $3 \pm 15$ Gt yr$^{-1}$. In all, the Antarctic Ice Sheet lost $2671 \pm 530$ Gt of ice between 1992 and 2020, raising
the global sea level by $7.4 \pm 1.5$ mm; after doubling in the mid-2000s from $62 \pm 41$ Gt yr$^{-1}$ to $130 \pm 45$ Gt yr$^{-1}$, increased
Antarctic ice losses – largely driven by an acceleration in ice discharge from the Amundsen Sea Sector (Mouginot et al., 2014)
– have persisted to the present-day. The rate of Greenland ice loss has remained highly variable during the last 5-year period
of our updated assessment, ranging from $86 \pm 75$ Gt yr$^{-1}$ in 2017 to a new maximum of $444 \pm 93$ Gt yr$^{-1}$ in 2019 driven by
exceptional surface melting during the summer (Tedesco and Fettweis, 2020). The majority of ice sheet losses have arisen
from Greenland during our 29-year survey: $4892 \pm 457$ Gt in total at an average rate of $169 \pm 16$ Gt yr$^{-1}$. Combined, Antarctica
and Greenland lost $7563 \pm 699$ Gt of ice between 1992 and 2020, raising the global sea level by $21 \pm 2$ mm.

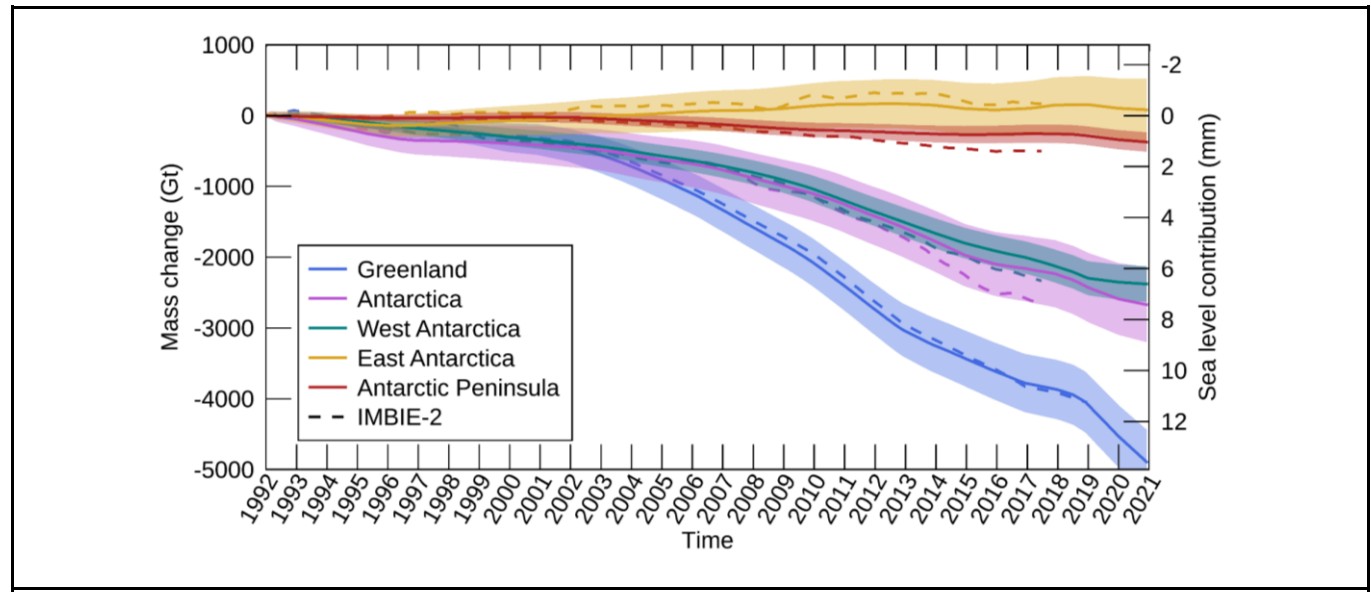

**Figure 4. Cumulative ice sheet mass changes. The estimated 1σ uncertainty of the cumulative change is shaded. The dashed lines show the results from our previous assessments (IMBIE-2).**

**Table 2. Rates of ice sheet mass change (Gt yr$^{-1}$). Rates are calculated from the first day (1$^{st}$ January) of the first year quoted to the last day (31$^{st}$ December) of the final year quoted in the table. The percentage in brackets is the fraction of sea level rise driven by the ice sheets (as the global mean sea level record starts in 1993, we do not compute the fraction of sea level rise from the ice sheets for the first time period of the table).**

|  | GrIS | AIS | WAIS | EAIS | APIS |
|---|---|---|---|---|---|
| **1992-1996** | -35 ± 29 | -70 ± 40 | -37 ± 19 | -27 ± 33 | -7 ± 11 |
| **1997-2001** | -48 ± 36 [4.0 %] | -19 ± 39 [1.6 %] | -42 ±19 [3.5 %] | 21 ± 32 [-1.7 %] | 2 ± 11 [-0.2 %] |
| **2002-2006** | -180 ± 39 [15.5 %] | -62 ± 41 [5.4 %] | -64 ± 20 [5.5 %] | 21 ± 34 [-1.8 %] | -20 ± 11 [1.7 %] |
| **2007-2011** | -280 ± 38 [31.8 %] | -130 ± 45 [14.8 %] | -129 ± 23 [14.6 %] | 19 ± 36 [-2.2 %] | -21 ± 12 [2.3 %] |
| **2012-2016** | -213 ± 40 [11.9 %] | -150 ± 43 [8.4 %] | -131 ± 21 [7.3 %] | -13 ± 35 [0.7 %] | -6 ± 13 [0.3 %] |
| **2017-2020** | -257 ± 42 [17.7 %] | -115 ± 55 [7.9 %] | -94 ± 25 [6.5 %] | 0 ± 47 [0 %] | -21 ± 12 [1.5 %] |
| **1992-2020** | -169 ± 16 [13.5 %] | -92 ± 18 [7.4 %] | -82 ± 9 [6.6 %] | 3 ± 15 [-0.2 %] | -13 ± 5 [1.0 %] |

## 5 Discussion

**5.1. Comparison to previous IMBIE assessment**

Finally, we assess the consistency of our results with our most recent assessment of ice sheet mass balance (IMBIE-2) to evaluate the impact of incorporating updated datasets and using an updated processing scheme. During their overlapping periods – 1992 to 2017 for Antarctica and 1992 to 2018 for Greenland – the results of this study and IMBIE-2 are in agreement within their respective uncertainties with rates of mass change of -150.0 ± 16 Gt yr$^{-1}$ and -150 ± 12 Gt yr$^{-1}$ for GrIS, respectively 355 and rates of -86 ± 19 Gt yr$^{-1}$ and -103 ± 22 Gt yr$^{-1}$ for AIS, respectively. Next, comparing rates of mass balance within calendar years shows that results from this study and our previous assessment are consistent across all years for all ice sheets, except for two years at the start of our record (1992 and 1995) at the GrIS for which the difference between our mass balance assessments exceeds their respective uncertainty bounds. On average, the magnitude of the differences in annual rates of mass balance is 36 Gt yr$^{-1}$ at GrIS, 33 Gt yr$^{-1}$ at AIS, 12 Gt yr$^{-1}$ at APIS, 31 Gt yr$^{-1}$ at EAIS, and 23 Gt yr$^{-1}$ at WAIS. The relatively 360 small differences between our previous and current mass balance assessments originate from a combination of our inclusion of updated datasets and the implementation of an updated processing scheme in this study. In all ice sheet regions, participant datasets have been updated compared to our previous assessment. In addition, in this study we apply a common processing

scheme to the AIS and GrIS, while in our previous study the mass balance assessments were aggregated with and without inverse-error weighting in the respective regions.


## 5.2 Comparisons to sea level contribution and projections of future sea level rise

Our assessment of ice sheet mass balance also provides a means of tracking the contribution of the ice sheets to GMSL. Here, we discuss the relative contributions of Greenland and Antarctica to GMSL by comparing our results to the GMSL trend from the AVISO product (https://www.aviso.altimetry.fr/msl/, last access: 12th April 2022). Although numerous satellite-altimetry-
based time-series of GMSL are available, differences between these products are less than 5 % of the GMSL trend (Ablain et al., 2019) and so the choice of one particular source does not affect our present discussion. From our updated assessment, Greenland and Antarctica have contributed $0.74 \pm 0.07$ mm yr$^{-1}$ to GMSL during the AVISO record (1993-2020), contributing 14 % and 8 % to the overall trend, respectively. This is consistent with findings from previous studies which examined the relative contributions of the different components of the sea level budget (WCRP, 2018; Horwath et al., 2022). Compared to
the pre-2000s period (1993-1999) when the ice sheets' contribution to GMSL was only $0.26 \pm 0.11$ mm yr$^{-1}$ (9 % of the GMSL trend), Greenland and Antarctica now (2010 to 2020) contribute $1.09 \pm 0.12$ mm yr$^{-1}$ (24 % of the GMSL trend) – four times higher. In particular, the acceleration of the ice sheets' contribution to GMSL was driven by increased ice losses from the GrIS (Chen et al., 2017; Dieng et al., 2017, Hamlington et al., 2020) with its contribution rising from $0.12 \pm 0.08$ mm yr$^{-1}$ pre-2000s to $0.68 \pm 0.08$ mm yr$^{-1}$ in the 2010s. In all periods post-2000, we find that the ice sheets make up at least 20 % of the GMSL
rise and during the period 2007-2011 in particular, ice losses accounted for 47 % of the GMSL rise due to accelerated ice losses from Greenland and West Antarctica during those 5 years (Table 2).

Satellite observations of ice sheet mass balance are important for evaluating ice sheet models and their climate model forcing (Shepherd and Nowicki, 2017; Slater et al., 2020; Aschwanden et al., 2021). In their 2021 assessment (AR6), the IPCC projected ice losses from Antarctica and Greenland due to SMB and glacier dynamics under a range of emission scenarios
every ten years, beginning in 2020 (Fox-Kemper et al., 2021) (Figure 4). As a result, we compare satellite mass balance rates from the decade prior (2010-2020) to those at the beginning of the projection period (2020-2030) (Table 3). In Antarctica, the observed sea level contribution during the last 10 years of our survey is $0.42 \pm 0.09$ mm yr$^{-1}$, closest to the median sea level contribution projected by the IPCC for the following decade (0.6 mm yr$^{-1}$). We note the large spread between the lower (10th percentile) and upper (90th percentile) ranges of the projected sea level contribution from Antarctica during this period –
between -0.1 mm yr$^{-1}$ and 2.2 mm yr$^{-1}$, respectively – even in their first decade. Although Greenland ice losses were highly variable between 2010 and 2020, they raised the global sea level at an average rate of $0.68 \pm 0.08$ mm yr$^{-1}$, closest to the median sea level contribution between 2020 and 2030 predicted by the IPCC (0.7 mm yr$^{-1}$). If the recent acceleration in Greenland ice losses were to continue ($1.2 \pm 0.2$ mm yr$^{-1}$ between 2019 and 2020), however, they would track above the upper

range predicted by the IPCC this decade (1.0 to 1.1 mm yr$^{-1}$ for all emission pathways). If ice sheet losses were to continue on
the median IPCC trajectory, the polar ice sheets will raise global sea levels by between 148 and 272 mm by 2100 (Figure 5).
Because the AR6 projections incorporate a long-term dynamic ice sheet response based on observations from the last 40 years
(Fox-Kemper et al., 2021), it follows that our assessment tracks closest to the median range in the near-term; as the overlap
period between our survey and AR6 predictions is only one year, a longer period of comparison is required to establish the
actual trajectory the ice sheets are following and the suitability of the time period used to assess the long-term dynamic
response. Remaining uncertainties in the Antarctic Ice Sheet response to climate forcing still drive the spread of climate model
projections, which range between -5 and 631 mm for both ice sheets at 2100.

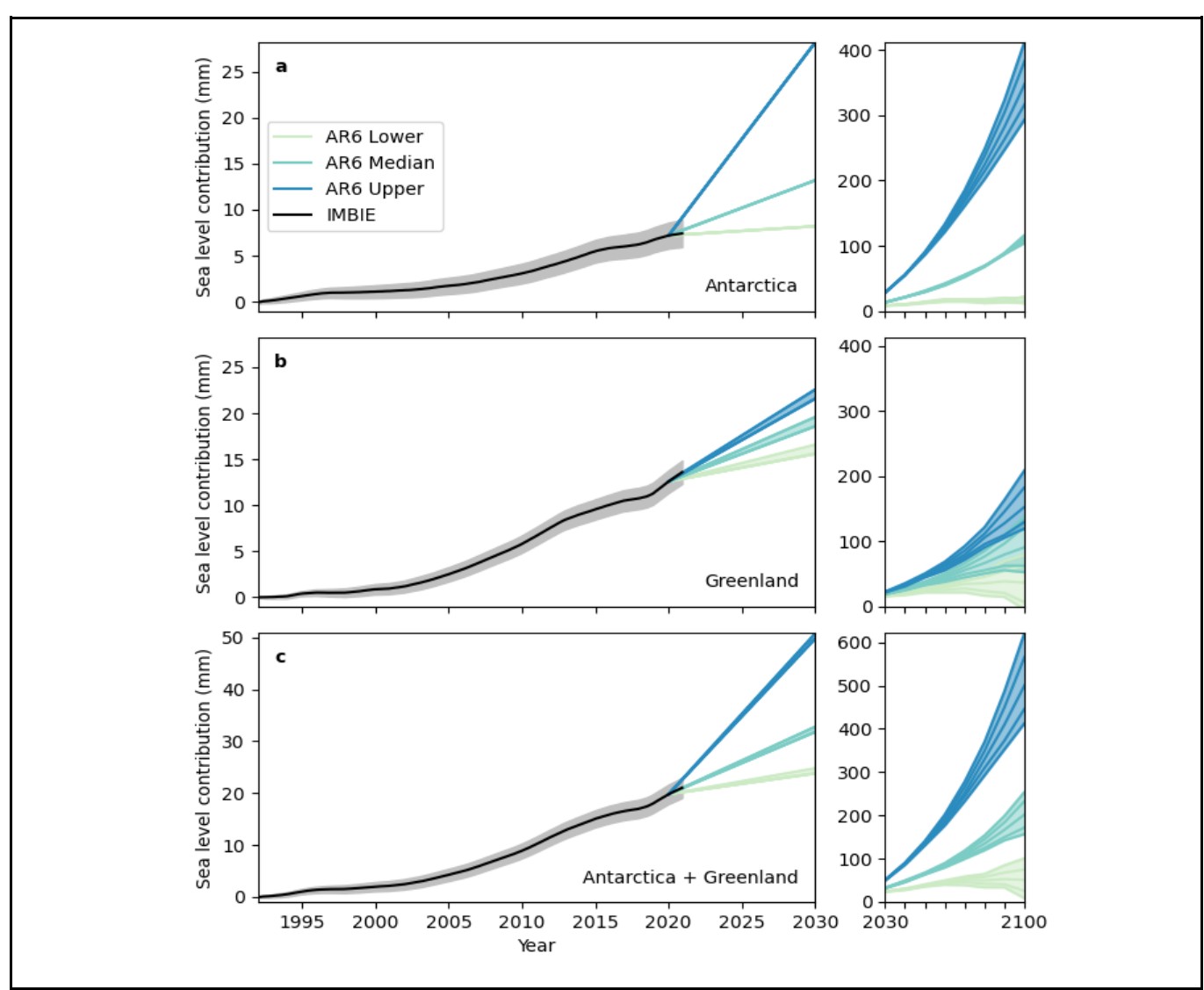

**Figure 5.Comparison of observed sea level contributions from a) the Antarctic Ice Sheet, b) Greenland Ice Sheet, c) Antarctic and Greenland Ice Sheets from this study (IMBIE) and predicted by the IPCC AR6 between 1992 and 2030 (left) and 2030 and 2100 (right). The AR6 upper, median and lower estimates are taken from the 90th percentile, median, and 10th percentile values of the ensemble range, respectively.**

| Table 3. Decadal rates of sea level contribution from IMBIE and AR6 projections | | | | |
|---|---|---|---|---|
| | 2010-2020 (mm yr$^{-1}$) | 2020-2030 (mm yr$^{-1}$) | | |
| | IMBIE | AR6 Lower | AR6 Median | AR6 Upper |
| AIS | 0.41 ± 0.09 | -0.1 - 0.0 | 0.6 | 2.1 - 2.2 |
| GrIS | 0.68 ± 0.08 | 0.4 - 0.5 | 0.7 | 1.0 - 1.1 |

### 5.3 Limitations of this study and roadmap for future improvements

In this section, we discuss the limitations of our dataset and a roadmap to improve ice sheet mass balance assessments. The inclusion of the peripheral glaciers and ice caps in the vicinity of the Greenland and Antarctic Ice Sheets is ambiguous in our assessment as not all individual estimates of ice sheet mass balance included here account for those. This relates to the varying ability of satellite techniques to resolve mass balance over those small glaciated areas. Space gravimetry has a coarse spatial resolution of a few hundred kilometres which is not sufficient to separate signals of mass change originating from the ice sheet

and its peripheral glaciers. On the other hand, the altimetry estimates included in this assessment exclude the peripheral glaciers and ice caps due to the complex terrain of these glaciers and their relatively small size compared to the footprint size of traditional pulse-limited altimeters. Finally, the input-output estimates do include mass changes from these glaciers, mostly by estimating their changes in SMB. Despite covering a relatively small area (around one tenth of the area of the ice sheets) (Pfeffer et al., 2014), these glaciers contribute significantly to global mean sea level rise with ice losses originating from the

Greenland and Antarctic Ice Sheets amounting to 36 ± 6 Gt yr$^{-1}$ and 21 ± 5 Gt yr$^{-1}$ during the period 2010-2019, respectively (Hugonnet et al., 2021). In addition, ice losses have accelerated in the periphery of the Greenland Ice Sheet, with glacier mass loss increasing by 64 % between 2003-2009 and 2018-2021 (Khan et al., 2022). These glaciers therefore need to be accounted for without ambiguity in future IMBIE assessments to remove systematic biases between the different satellite techniques linked to their (non-)inclusion in individual mass balance estimates. Recent progress in satellite altimetry, with the

development of CryoSat-2 swath radar altimetry for measuring mass changes of mountain glaciers (Foresta et al., 2016; Jakob et al., 2021) and the launch of ICESat-2, already contribute to a better mapping of those glaciers. New community initiatives, such as GlamBIE (the Glacier mass balance Inter-comparison Exercise), will further contribute to separating mass changes between the ice sheets and glaciers lying at their periphery by offering a consensus-estimate that could be removed from the gravimetry estimates that currently account for both.

Continuing efforts to understand the remaining differences between altimetry, gravimetry, and the input-output method is critical to provide more robust observational estimates of the contribution of the ice sheets to GMSL. Producing estimates with a better temporal resolution by using data from the newest satellite missions, reprocessing the satellite record with the newest geophysical corrections, and using a better uncertainty characterisation, will undoubtedly help further reconcile satellite assessments of ice sheet mass balance produced from different techniques. To achieve this, it is also important to assess the
impact of SMB and GIA models. SMB processes are responsible for a large proportion of Greenland's ice losses (and to a lesser extent of Antarctica's ice losses) (Enderlin et al., 2014; Shepherd et al., 2020), and thus pursuing the efforts of recent model inter-comparisons (Fettweis et al., 2020; Mottram et al., 2021) is key to improve the agreement between input-output estimates but also to partition mass trends into SMB and ice dynamics components as it provides critical information on the dominant processes at play. A model-inter-comparison of GIA models would also be timely as new approaches have been
developed in recent years to determine the GIA signal (Whitehouse, 2018). New data-driven solutions that rely on present day geodetic observations (e.g. Riva et al., 2009; Vishwakarma et al., 2022), and solutions derived from coupling a GIA model to an ice sheet mode (de Boer et al., 2017) have become available. Examining the variability of GIA solutions determined from forward models, data inversion, and coupled models will help reducing uncertainties in space gravimetry estimates of ice sheet mass balance.

Finally, improving the spatial resolution of the IMBIE assessment by producing time-series of mass changes within the individual basins of the Greenland and Antarctic Ice Sheets will also contribute to further identify areas of similarities and disagreement between satellite techniques (Sutterley et al, 2014) and will support the identification of spatial biases in satellite estimates of ice sheet mass balance. In addition, regional assessments of ice sheet mass balance could support the evaluation and calibration of ice sheet models, contributing to reducing uncertainties in future sea level rise projections (Edwards et al.,
2021; Nias et al., 2019).

**6 Conclusions**

We combine 50 estimates of ice sheet mass balance, 26 for Greenland and 24 for Antarctica, to produce a new reconciled estimate of ice sheet mass balance showing that the ice sheets lost 7,563 ± 699 Gt of ice between 1992 and 2020. Ice losses have accelerated at both ice sheets over this 29-year record and the rate of ice loss is now 5 times higher in Greenland and 25
% higher in Antarctica compared to the early 1990s. Our assessment shows that the altimetry, gravimetry, and input-output

method are in close agreement in Greenland with a spread of 19 Gt yr$^{-1}$ over their common time period, which represents only 10.9 % of the rate of imbalance. In Antarctica, the spread between techniques is 4 times larger than in Greenland, mostly due to large differences between estimates for the East Antarctic Ice Sheet. To further explore and interpret differences between geodetic techniques, producing altimetry estimates with a higher temporal resolution (especially during the first half of the satellite altimetry record), better GIA constraints for the gravimetry estimates, and additional estimates of ice sheet mass balance via the input-output method would improve the comparison and aggregation of ice sheet mass balance estimates. Continuously monitoring the mass balance of the ice sheets and producing annual updates of Greenland and Antarctica mass balance is critical to track their contribution to global mean sea level and constrain projections of future sea-level rise.

## 7 Data Availability

The aggregated Greenland and Antarctic Ice Sheets mass balance data and associated errors generated in this study are freely available at the NERC Polar Data Centre, https://doi.org/10.5285/77B64C55-7166-4A06-9DEF-2E400398E452 (The IMBIE Team, 2021).

## 8 Code Availability

The code used to compute and aggregate rates of ice sheet mass change and their errors is freely available at https://github.com/IMBIE.

## Author Contribution

The executive committee of IMBIE (A.S., E.R.I., N.J.S., I..N.O, C.A., M.B., M.H.,. I.J., M.D.K., G.K., S.N., A.J.P., E.R., T.S., K.M.S., B.E.S., L.S.S., I.V., P.L.W.) designed the study. A.S., L.S.S., A.G., L.G., N.G., B.C.G., V.H., S.A.K, H.K., M.M, J.N., N.P., L.S., and S.B.S. contributed altimetry estimates. M.H., I.V., A.B., R.F., A.G., A. Groh, C.H., C.L., B.D.L., J.N., I.S., H.V.S., K.S., E.J.O.S., T.C.S., B.D.V, D.W., and B.W contributed gravimetry estimates. E.R., A.P.A., W.C., J.M., and B.S. contributed input-output estimates. M.B., C.Agosta, X.F., H.G., C.K., P.L.L., S.H.M., R.M., and B.N. contributed SMB estimates. G.A., B.S.L, D.M., W.R.P, G.S., M.W., and W.W. contributed GIA estimates. I.N.O. and M.E.P. performed the mass balance data collation. I.N.O. prepared the datasets comparison. T. Slater performed the AR6 data analysis and prepared Fig 4 and Table 3. I.N.O. led the writing and prepared the other figures and tables. I.N.O, A.S. and T. Slater wrote the manuscript. All authors participated in the data interpretation and commented on the manuscript.

## Competing interests

C. A. is member of the editorial board of journal Earth System Science Data.

**Acknowledgements**

This work is an outcome of the Ice Sheet Mass Balance Inter-comparison Exercise (IMBIE) supported by the ESA EOEP-5
'EO Science for Society', the ESA 'Climate Change Initiative', and the NASA Cryosphere Program. Research was carried out
at the Jet Propulsion Laboratory, California Institute of Technology, under a contract with the National Aeronautics and Space
Administration (NASA). Funding for E.I. and N-J.S. was provided by NASA ROSES solicitation NNH20ZDA001N-CRYO
in response to Proposal 20-CRYO2020-0003. GEUS data provided from the Programme for Monitoring of the Greenland Ice
Sheet (www.PROMICE.dk) was funded by the Danish Ministry of Climate, Energy and Utilities. M.M. acknowledges the
support of the UK NERC Centre for Polar Observation and Modelling (CPOM), and the Lancaster University-UKCEH Centre
of Excellence in Environmental Data Science. P. L. L. gratefully acknowledges the contributions of Aarhus University
Interdisciplinary Centre for Climate Change (iClimate, Aarhus University). N.G. used CryoSat data obtained from ESA at
cs2eo.org and via the CryoTEMPO-EOLIS project https://cryotempo-eolis.org/. G.S. is funded by a research grant of DIFA
(Dipartimento di Fisica e Astronomia "Augusto Righi") of the Alma Mater Studiorum Universita` di Bologna. J.N. and A.G.
were supported by the ITS_LIVE project awarded through NASA MEaSUREs program,
and the NASA Cryosphere program through participation in the
ICESat-2 science team. I.S. acknowledges funding by the Helmholtz Climate Initiative REKLIM (Regional Climate Change),
a joint research project of the Helmholtz Association of German Research Centres (HGF). Ice velocity data for Greenland and
Antarctica provided by UC Irvine is funded by NASA MEaSUREs program. BedMachine Antarctica is funded by NASA
MEaSUREs program. BedMachine Greenland is funded by research grants from NASA Operation IceBridge Mission. BW
was funded by NWO VIDI grant 016.Vidi.171.063.

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

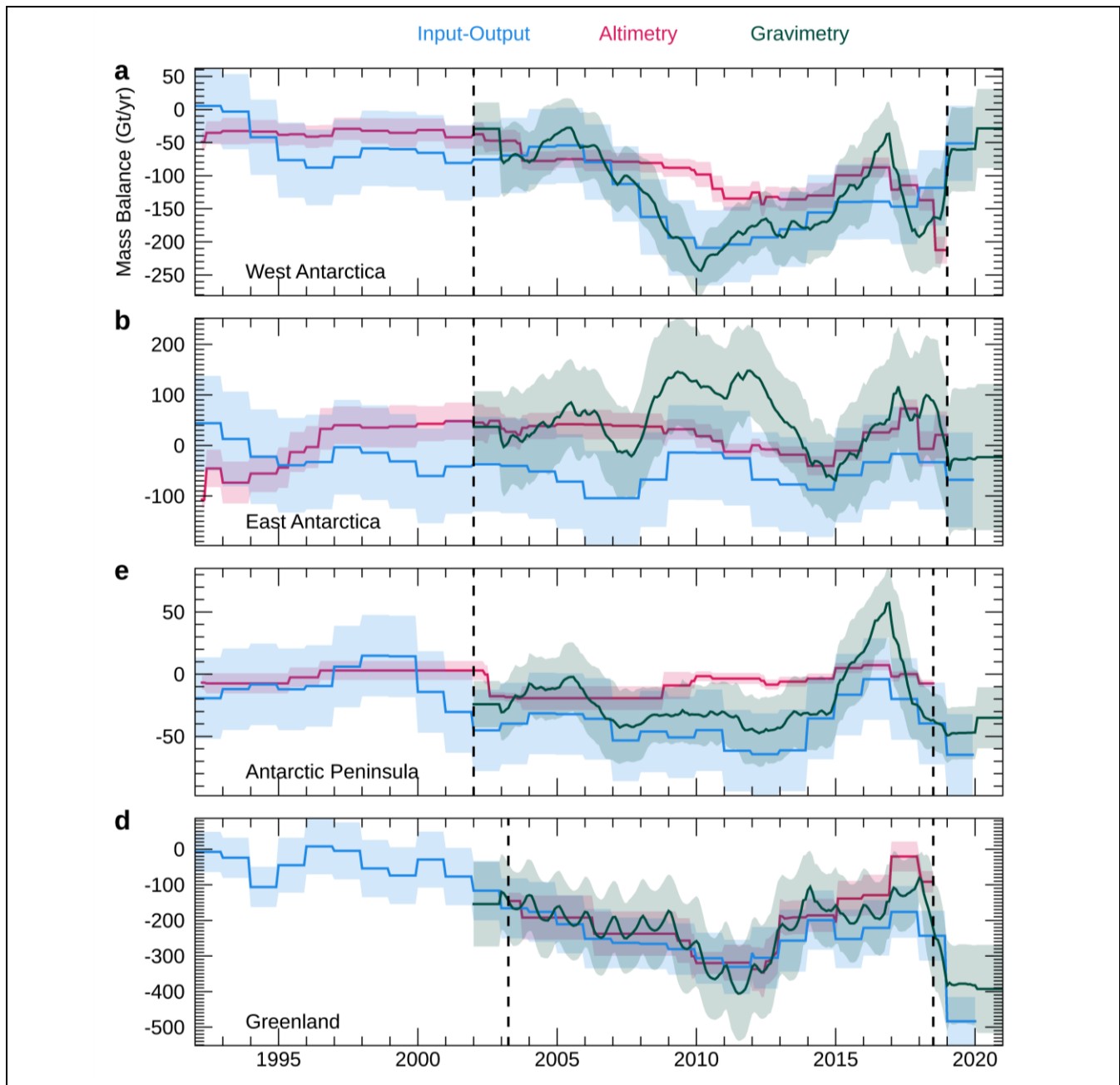

**Figure A1. Mass balance time-series from the aggregated altimetry, gravimetry and input-output method over the a) WAIS, b) EAIS, c) APIS, and d) GrIS. The vertical dashed lines mark the overlap period of the three time-series.**

**Table A1. References of the datasets, methods, GIA and SMB models employed by participants of the input-output, altimetry and gravimetry experiment groups.**

| | | |
|---|---|---|
| **IOM** | Andersen | Andersen, M. L. *et al.* Basin-scale partitioning of Greenland ice sheet mass balance components (2007–2011). *Earth and Planetary Science Letters* **409**, 89–95 (2015). |
| | Colgan | Colgan, W. *et al.* Greenland ice sheet mass balance assessed by PROMICE (1995–2015). *Geological Survey of Denmark and Greenland Bulletin* **43** (2019). |
| | Mouginot | Mouginot, J. *et al.* Forty-six years of Greenland Ice Sheet mass balance from 1972 to 2018. *PNAS* **116**, 9239–9244 (2019). |
| | Rignot | Rignot E. *et al.* Four decades of Antarctic Ice Sheet mass balance from 1979-2017. *PNAS* **116**(4), 1095-1103 (2019). |
| **ALT** | Gourmelen | Gourmelen, N. *et al.* CryoSat-2 swath interferometric altimetry for mapping ice elevation and elevation change. *Advances in Space Research* **62**, 1226–1242 (2018). |
| | Gunter | Gunter, B. C. *et al.* Empirical estimation of present-day Antarctic glacial isostatic adjustment and ice mass change. *The Cryosphere* **8**, 743–760 (2014). |
| | Helm | Helm, V., Humbert, A. & Miller, H. Elevation and elevation change of Greenland and Antarctica derived from CryoSat-2. *The Cryosphere* **8**, 1539–1559 (2014). |
| | Khan | Khan, S. A. et al. Sustained mass loss of the northeast Greenland ice sheet triggered by regional warming. Nature Climate Change 4, 292–299 (2014). |
| | McMillan | McMillan, M. *et al.* A high-resolution record of Greenland mass balance. *Geophysical Research Letters* **43**, 7002–7010 (2016). |
| | Nilsson Gardner | Nilsson, J., Gardner, A., Sandberg Sørensen, L. & Forsberg, R. Improved retrieval of land ice topography from CryoSat-2 data and its impact for volume-change estimation of the Greenland Ice Sheet. *The Cryosphere* **10**, 2953–2969 (2016). |
| | Pie | Felikson, D. *et al*. Comparison of Elevation Change Detection Methods From ICESat Altimetry Over the Greenland Ice Sheet. *IEEE Transactions on Geoscience and Remote Sensing* **55**, 5494–5505 (2017). |

| | | |
|---|---|---|
| | Sandberg Sørensen | Sørensen, L. S. *et al.* Mass balance of the Greenland ice sheet (2003–2008) from ICESat data – the impact of interpolation, sampling and firn density. *The Cryosphere* **5**, 173–186 (2011). |
| | Schroder | Schröder, L. *et al.* Four decades of Antarctic surface elevation changes from multi-mission satellite altimetry." *The Cryosphere* **13**(2), 427-449, (2019). |
| | Shepherd | Shepherd, A. *et al.* Trends in Antarctic Ice Sheet Elevation and Mass. *Geophysical Research Letters* **46**(14), 8174-8183 (2019). |
| | Zwally | Zwally, H. J. *et al.* Mass gains of the Antarctic ice sheet exceed losses. *J. Glaciol.* **61**, 1019-1036 (2015). |
| **GMB** | Blazquez | Blazquez, A. *et al.* Exploring the uncertainty in GRACE estimates of the mass redistributions at the Earth surface: implications for the global water and sea level budgets. *Geophys J Int* **215**, 415–430 (2018). |
| | Bonin | Bonin, J. & Chambers, D. Uncertainty estimates of a GRACE inversion modelling technique over Greenland using a simulation. *Geophys J Int* **194**, 212–229 (2013). |
| | Forsberg | Forsberg, R., Sørensen, L. & Simonsen, S. Greenland and Antarctica Ice Sheet Mass Changes and Effects on Global Sea Level. *Surv Geophys* **38**, 89–104 (2017). |
| | Gardner Nilsson | Gardner, A. S. *et al.* Increased West Antarctic and unchanged East Antarctic ice discharge over the last 7 years. *The Cryosphere* **12**(2), 521-547 (2018). |
| | Groh | Groh, A., & Horwath, M. Antarctic Ice Mass Change Products from GRACE/GRACE-FO Using Tailored Sensitivity Kernels. Remote Sensing, 13, 1736, https://doi.org/10.3390/rs13091736 (2021). |
| | Harig | Harig, C. & Simons, F. J. Mapping Greenland's mass loss in space and time. *PNAS* **109**, 19934–19937 (2012). |
| | Horvath | Horvath, A. G. Retrieving Geophysical Signals from Current and Future Satellite Missions. PhD thesis, Tech. Univ. Munich (2017). |
| | Luthcke | Luthcke, S. B. *et al.* Antarctica, Greenland and Gulf of Alaska land-ice evolution from an iterated GRACE global mascon solution. *Journal of Glaciology* **59**, 613–631 (2013). |

| | Moore | Andrews, S. B., Moore, P. & King, M. A. Mass change from GRACE: a simulated comparison of Level-1B analysis techniques. *Geophys J Int* **200**, 503–518 (2015). |
|---|---|---|
| | Save | Save, H., Bettadpur, S. & Tapley, B. D. High-resolution CSR GRACE RL05 mascons. *Journal of Geophysical Research: Solid Earth* **121**, 7547–7569 (2016). |
| | Schrama | Schrama, E. J. O., Wouters, B. & Rietbroek, R. A mascon approach to assess ice sheet and glacier mass balances and their uncertainties from GRACE data. *Journal of Geophysical Research: Solid Earth* **119**, 6048–6066 (2014). |
| | Seo | Seo, K.-W. *et al.* Surface mass balance contributions to acceleration of Antarctic ice mass loss during 2003–2013. *Journal of Geophysical Research: Solid Earth* **120**, 3617–3627 (2015). |
| | Velicogna | Velicogna, I., Sutterley, T. C. & Broeke, M. R. van den. Regional acceleration in ice mass loss from Greenland and Antarctica using GRACE time-variable gravity data. *Geophysical Research Letters* **41**, 8130–8137 (2014). |
| | Vishwakarma | Vishwakarma, B. D., Horwath, M., Devaraju, B., Groh, A. & Sneeuw, N. A Data-Driven Approach for Repairing the Hydrological Catchment Signal Damage Due to Filtering of GRACE Products. *Water Resources Research* **53**, 9824–9844 (2017). |
| | Wiese | Wiese, D. N., Landerer, F. W. & Watkins, M. M. Quantifying and reducing leakage errors in the JPL RL05M GRACE mascon solution. *Water Resources Research* **52**, 7490–7502 (2016). |
| | Wouters | Wouters, B., Bamber, J. L., van den Broeke, M. R., Lenaerts, J. T. M. & Sasgen, I. Limits in detecting acceleration of ice sheet mass loss due to climate variability. *Nature Geoscience* **6**, 613–616 (2013). |
| **GIA** | A13 | A, Geruo, John Wahr, Shijie Zhong, Computations of the viscoelastic response of a 3-D compressible Earth to surface loading: an application to Glacial Isostatic Adjustment in Antarctica and Canada, *Geophysical Journal International*, Volume 192, Issue 2, February 2013, Pages 557–572, https://doi.org/10.1093/gji/ggs030 |
| | W12a | Whitehouse, Pippa L., Michael J. Bentley, Glenn A. Milne, Matt A. King, Ian D. Thomas, A new glacial isostatic adjustment model for Antarctica: calibrated and tested using observations of relative sea-level change and present-day uplift rates, *Geophysical Journal International*, |

| | | Volume 190, Issue 3, September 2012, Pages 1464–1482, https://doi.org/10.1111/j.1365-246X.2012.05557.x |
|---|---|---|
| | ICE-5G | Peltier, W. R. (2004). "Global Glacial Isostasy And The Surface Of The Ice-Age Earth: The ICE-5G (VM2) Model and GRACE." Annual Review of Earth and Planetary Sciences **32**(1): 111-149. |
| | ICE-6G | Peltier, W. R., Argus, D. F., & Drummond, R. (2015). Space geodesy constrains ice age terminal deglaciation: The global ICE-6G_C (VM5a) model. *Journal of Geophysical Research: Solid Earth*, *120*(1), 450-487. |
| | IJ05 | Ivins, E., & James, T. (2005). Antarctic glacial isostatic adjustment: A new assessment. *Antarctic Science, 17*(4), 541-553. doi:10.1017/S095410200500296) |
| | IJ05_R2 | Ivins, E. R., T. S. James, J. Wahr, E. J.O. Schrama, F.W. Landerer & K. M. Simon, Antarctic contribution to sea-level rise observed by GRACE with improved GIA correction, *J. Geophys. Res., B - Solid Earth*, 118, 3126-3141, doi:10.1002/jgrb.50208 (2013). |
| | Paulson07 | Paulson, A., Zhong, S. and Wahr, J. (2007), Inference of mantle viscosity from GRACE and relative sea level data. Geophysical Journal International, 171: 497-508. https://doi.org/10.1111/j.1365-246X.2007.03556.x |
| | Simpson09 | Simpson, M. J. R. and Milne, G. A. and Huybrechts, P. and Long, A. J. (2009) 'Calibrating a glaciological model of the Greenland ice sheet from the last glacial maximum to present-day using field observations of relative sea level and ice extent.', *Quaternary science reviews.*, 28 (17-18). pp. 1631-1657. |
| | Khan_2016 | Khan, S. A., I. Sasgen, M. Bevis, T. v. Dam, J. L. Bamber, J. Wahr, M. Willis, K. H. Kjær, B. Wouters, V. Helm, B. Csatho, K. Fleming, A. A. Bjørk, A. Aschwanden, P. Knudsen and P. K. Munneke (2016). "Geodetic measurements reveal similarities between post-Last Glacial Maximum and present-day mass loss from the Greenland ice sheet." *Science Advances* **2**(9): e1600931. |
| | Schrama14 | Schrama, E. J. O., Wouters, B., and Rietbroek, R. (2014), A mascon approach to assess ice sheet and glacier mass balances and their uncertainties from GRACE data, *J. Geophys. Res. Solid Earth*, 119, 6048– 6066, doi:10.1002/2013JB010923. |

| | | |
|---|---|---|
| **SMB** | RACMO 2.3 | Van Wessem, J. M., Reijmer, C. H., Morlighem, M., Mouginot, J., Rignot, E., Medley, B., Joughin, I., Wouters, B., Depoorter, M. A., Bamber, J. L., Lenaerts, J. T. M., Van De Berg, W. J., Van Den Broeke, M. R. and Van Meijgaard, E. (2014) "Improved representation of East Antarctic surface mass balance in a regional atmospheric climate model," *Journal of Glaciology*, Cambridge University Press, 60(222), pp. 761–770. |
| | MAR 3.5 | Fettweis, X., B. Franco, M. Tedesco, J. H. van Angelen, J. T. M. Lenaerts, M. R. van den Broeke and H. Gallée (2013). "Estimating the Greenland ice sheet surface mass balance contribution to future sea level rise using the regional atmospheric climate model MAR." The Cryosphere **7**(2): 469-489. |

860