# Peer review of "Mass Balance of the Greenland and Antarctic Ice Sheets from 1992 to 2020"

_Earth System Science Data, 2022_

## Author Comment (AC1)

We would like to thank Anny Cazenave and Ellyn Enderlin for their reviewer comments and Ken Mankoff, Romain Hugonnet and Etienne Berthier for their community comments and suggestions. In this response we address the concerns raised by the reviewers and the community. Please find our responses in blue and the changes that we implemented to address those comments in green below the comments.

**Response to Reviewer Comment #1 (Anny Cazenave)**

This study is an update of the previous IMBIE assessments of the Greenland and Antarctica mass balances based on different space-based estimates provided by several groups worldwide. The new time series of combined ice sheet mass balances are extremely useful for the community, in particular for scientists interested in studying the global mean sea level budget. The paper is clearly written and should be published after accounting for a few minor corrections.

Thank you very much for your positive review and for highlighting the usefulness of our dataset to the scientific community!

**RC1.1.** My main comment concerns the systematic differences reported by the authors between the three methods used for estimating the Greenland and Antarctica mass balances (IOM, altimetry and GRACE) as well as on the solutions dispersion within each method. As shown in the present study, satellite altimetry provides more dispersed solutions (lines 240-241) than the other two methods, while the IOM approach leads to systematically lower estimates than altimetry and space gravimetry (Fig.2). The first IMBIE assessment was published 10 years ago and I am sure that the authors have investigated the reasons for such discrepancies. I thus recommend that a discussion be added in the present paper on the potential causes of the reported dispersion of altimetry solutions and of the systematic discrepancies between the 3 methods. A few words on perspectives to reduce them in the future (if possible) would also be welcome. I would also suggest that you show (e.g., in a Supplementary Material section) the different mass balance time series for each method separately (not only annual rates estimates as in Appendix A).

**Response:** Thank you for this suggestion. We have added a new subsection in our discussion to outline the current limitations of our dataset (especially with respect to the inclusion/exclusion of the peripheral glaciers and ice caps) and a roadmap for future improvements. We discuss future work that we are planning on carrying out in future IMBIE assessments that will contribute to better understand the remaining discrepancies between altimetry, gravimetry and input-output estimates. This includes performing inter-comparisons of SMB and GIA models to better assess the impact of using different models on input-output and gravimetry estimates; partitioning mass trends into their SMB and dynamics components; and improving the spatial resolution of the IMBIE assessments by producing reconciled time-series of mass changes within the individual basins of the Greenland and Antarctic Ice Sheets to further identify areas of similarities and disagreement between satellite techniques.
We have also added a supplementary figure showing the mass balance time-series for each method separately as suggested.

**Actions:**
- We added a new subsection '5.3 Limitations of this study and roadmap for future improvements' in our discussion of the results to outline the current limitations of our dataset and some recommendations for further reconciling estimates of ice sheet mass balance
- We added a supplementary figure (Figure A1) showing the mass change time-series for each method separately and for each ice sheet.

Minor comments:

-In the abstract, ice mass loss values are either positive or negative. Please use the same sign for all

**Response:** We have updated the abstract so that all values are now positive with the text stating if it corresponds to a mass loss or gain.

-Lines 89 to 100: for non experts, explain what is the GIA correction and how it affects each method

**Response:** Agreed, we have added an explanation of the GIA correction and how it affects the gravimetry and altimetry mass balance estimates.

**Action:** We added the following text: 'The GIA is the result of solid Earth mass redistribution caused by changes in ice mass since the last glaciation. Gravimetry fields record the combined effect of mass redistribution due to the GIA and recent changes in ice sheet mass balance. The GIA contribution therefore needs to be modelled separately and removed from the gravimetry fields, especially since it is of the same order of magnitude as the ice sheet mass balance signal (Caron and Ivins, 2020; Sutterley et al., 2014a). Altimetry elevation change estimates also need to be corrected for the GIA. However, contrary to gravimetry estimates, altimetry estimates are less sensitive to GIA as it manifests as an uplift (or subsidence) rate of the order of a few millimetres per year, much smaller than the elevation changes recorded.'

Line 93: quote GRACE after 'space gravimetry'

**Response:** Added.

**Action:** We corrected the corresponding text as: '[..] it is now possible to routinely monitor ice sheet mass changes using observations of [..] fluctuations in Earth's gravity field (derived from satellite gravimetry from GRACE and its follow on)'

Line 126: clarify the sentence '...orbit crossing' (e.g., difference in ice sheet elevation at a crossover point between ascending and descending satellite passes)

**Response:** Added, thank you for the suggestion.

**Action:** We modified the text as suggested: 'Satellite altimetry measures ice sheet elevation change, computed at orbit crossing points by calculating the difference in ice sheet elevation at a crossover point between ascending and descending satellite passes'

Line 143: Quote land hydrology when refering to leakage of mass trends in the climate system

**Response:** Added.

**Action:** We have amended this sentence as: 'Ice sheet mass changes are determined after making model-based corrections for GIA (e.g. Caron and Ivins, 2020) and for the leakage of mass trends occurring elsewhere in the climate system, especially those arising from ocean mass variability and changes in land hydrology.'

Lines 145 to 149: it seems that you use the words 'satellite gravimetry when you refer to GRACE and GRACE FO when you refer to GRACE Follow On. Space gravimetry' is the generic term. Indice more clearly is all 'space gravimetry' estimates include GRACE FO

**Response:** Agreed, we have corrected this.

**Action:** We have corrected the corresponding text as: 'This set of updated estimates is made of 2 estimates from the input-output method, 1 altimetry estimate, and 9 gravimetry estimates including data from the new GRACE Follow-On space gravimetry mission (GRACE-FO).'

Fig.4: The figure caption is quite brief and not fully clear. Indicate that the starting points of the curves shown in the right hand side panels are the 2030 values of the left hand side panels

**Response:** We have amended the caption accordingly.

**Action:** We have updated the caption as: 'Comparison of observed sea level contributions from a) the Antarctic Ice Sheet, b) Greenland Ice Sheet, c) Antarctic and Greenland Ice Sheets from this study (IMBIE) and predicted by the IPCC AR6 between 1992 and 2030 (left) and 2030 and 2100 (right). The AR6 upper, median and lower estimates are taken from the 90th percentile, median, and 10th percentile values of the ensemble range, respectively.'

**Response to Reviewer Comment #2 (Ellyn Enderlin)**

**Summary**

The paper describes the process by which 50 mass balance time series for the ice sheets were combined to produce a consensus estimate of ice sheet mass loss since the early 1990s and then summarizes the results and their implications. They find that the scientific community is in generally good agreement regarding rates of mass loss within and across the three different methodologies used for these estimates – altimetry, input-output, and gravimetry – with the largest disagreements for the East Antarctic Ice Sheet. The paper is concise and generally well-written with several summary tables and figures that aid the presentation. I appreciate the complex data wrangling that likely took place to produce this paper and thank the authors for producing an updated IMBIE dataset. I have a few major comments regarding the presentation of numbers throughout the text as well as some minor figure recommendations, as described below.

Thank you very much for your review and suggestions.

**Major Comments**

**RC2.1** At the end of the Data section, two different basin definitions are described but then it isn't clear how these are used in the analysis. The data all seem to be split according to GrIS, APIS, WAIS, and EAIS, not these smaller drainage basins. When are these different basins used? The basin use should be clarified in the Methods section.

**Response:** We reconcile mass trends over the GrIS, APIS, WAIS and APIS and not over the smaller drainage basins. We have clarified this in the revised manuscript. We also now discuss improvements to future IMBIE assessments in Section 5, including the possibility of providing reconciled time-series within ice sheet basins and discuss the remaining challenges to overcome to achieve this.

**Actions:**
- We split the 'Data' section into 'Input Data' and 'Output Data' (following Ken Mankoff's suggestion). In the 'Input Data' section, we added 'IMBIE participants were free to use either of these two definitions, and we combine mass trends over the GrIS, AIS, WAIS, EAIS, and APIS together regardless of what definition was chosen', and we further clarified in the 'Output Data' section that we reconcile mass trends over the main ice sheets only 'The output data consists of a single reconciled estimate of ice sheet mass balance covering the period 1st January 1992 to 31st December 2020 for the GrIS, AIS, APIS, WAIS, EAIS, and the sum of the GrIS and AIS.'
- We added a short paragraph in the discussion on reconciling mass balance estimates within the individual basins of the Greenland and Antarctic Ice Sheets: 'Finally, improving the spatial resolution of the IMBIE assessment by producing time-series of mass changes within the individual basins of the Greenland and Antarctic Ice Sheets will also contribute to further identify areas of similarities and disagreement between satellite techniques (Sutterley et al, 2014) and will support the identification of spatial biases in satellite estimates of ice sheet mass balance. In addition, regional assessments of ice sheet mass balance could support the evaluation and calibration of ice sheet models, contributing to reducing uncertainties in future sea level rise projections (Edwards et al., 2021; Nias et al., 2019).'

**RC2.2**. In the dataset descriptions and the results, you state that the input-output method provides annual temporal resolution but in the methods your explanation of the dataset integration describes all estimates as monthly. Are the input-output datasets monthly? Do they have regular temporal intervals? It would be helpful to add temporal sampling flag or some other indicator of temporal resolution to Table 1. Similarly, these datasets are described as all relying on the same SMB model. That model should be explicitly stated since SMB is a tremendously important component of GrIS mass loss.

**Response:** The input-output estimates included in this assessment are all posted at annual resolution. We resample those over regular monthly epochs to integrate them with the altimetry and gravimetry estimates. (In the next IMBIE assessment that we are currently preparing, we received new input-output estimates with finer temporal resolution.) We have revised Paragraph 1 of the Results section to clarify this and added a couple of sentences discussing the temporal resolution of the input-output method in the introduction. However, we did not add an extra row in Table 1 to indicate the temporal sampling of the different estimates as in this table, we summarise the main satellite missions and corrections used per technique group and not per estimate.

On the SMB models used, we have corrected the text and now explicitly state in the text the names of two SMB models used in the input-output estimates included (MAR and RACMO) in Greenland.

**Actions:**

**-** We revised Paragraph 1 of the Results section on the temporal resolution of the input-output method: 'During the last decade, new satellite missions with a more frequent revisit time (down to 6 days using image pairs from Sentinel-1a and Sentinel-1b available during the period 2016 to 2021 until the end of Sentinel-1b mission) have been used to improve the temporal resolution of ice velocity measurements, allowing to investigate seasonal fluctuations in ice velocity (King et al., 2018; Lemos et al., 2018) and produce monthly estimates of ice discharge at the continental scale.'

**-** In the 'Input Data' section, we added the names of the two models used in the input-output estimates included: 'In this assessment, only two SMB models have been used in the input-output method estimates included – the RACMO (Regional Atmospheric Climate Model) and MAR (Modèle Atmosphérique Régional) models (Table 1).

- In the first paragraph of the Results section, we have added the following sentences: 'To estimate the SMB anomaly in Greenland, two estimates used MAR (version 3.2 and version 3.5.2) and one used RACMO (version 2.3). In Antarctica, the input-output estimate used RACMO (version 2.3). In addition to using different SMB models, those estimates also define different reference periods to calculate the SMB anomalies. All of the mass balance estimates derived in this group were originally posted at annual resolution and we resample them over monthly epochs to aggregate them with estimates from the other groups.'

**-** In the Discussion section, we added a sentence on the importance of SMB processes on ice sheet mass balance: 'SMB processes are responsible for a large proportion of Greenland's ice losses (and to a lesser extent of Antarctica's ice losses) (Enderlin et al., 2014; Shepherd et al., 2020), and thus pursuing the efforts of recent model inter-comparisons (Fettweis et al., 2020; Mottram et al., 2021) is key to improve the agreement between input-output estimates but also to partition mass trends into SMB and ice dynamics components as it provides critical information on the dominant processes at play.'

**RC2.3.** Throughout the results, I was uncertain how to interpret some of the metrics presented as summaries for the datasets and their intercomparison. It seems like the maximum difference in datasets is often reported. Why is this used and not the median or the trend? Why report the average of the standard deviations of the datasets? For small sample sizes, the average may be highly skewed. Finally, what are the metrics presented for the aggregate datasets? Are they the mean +/- standard deviation? Is the standard deviation calculated using the standard deviations of the independent datasets or is it a metric of variability over time for the aggregate dataset?

**Response:** We have modified this in the revised version of the paper as suggested and now state the median and standard deviation of the differences instead of the maximum difference. In the original manuscript, we calculated the standard deviation of the different datasets in each year and reported the mean of those annual standard deviations. We have updated this in the revised manuscript and now report the median difference and standard deviation (not averaged).

When comparing the aggregated datasets together, we report the standard deviation of the aggregated-altimetry, gravimetry and input-output estimates rates of mass change during their overlap period over the different regions and compare it to the reconciled rate of mass change and its uncertainty (computed as described in section 3). The standard deviation calculated is thus the standard deviation between the

mean rate of mass change of the three independent estimates (the three aggregated altimetry, gravimetry, input-output time-series) and not a metric of the variability in over time. We have clarified this in the text.

**Actions:**
-    We updated all metrics in paragraphs 1 to 4 of the Results section to report the median difference and standard deviation instead of the maximum difference and averaged standard deviation.
-    We clarified in paragraph 5 of the Results section the metrics we use for the aggregate datasets: 'We report the standard deviation of the aggregated-altimetry, gravimetry and input-output estimates rates of mass change and compare it to the reconciled rate of mass change and its uncertainty (computed as described in Section 3).'

**RC2.4**. At the beginning of the discussion, the aggregate rates of mass loss are compared to trends in global sea level rise. In addition to their contribution to the trend, it would be helpful to know what fraction of annual sea level rise is driven by ice sheet mass loss.

**Response:** Thank you for the suggestion, we have added in Table 2 the fraction of sea level rise driven by the ice sheets over 5-year intervals of our record using the same time intervals as for the rates of mass change presented in the table. (The GMSL time-series starts in 1993, so we only computed the fraction over the intervals 1997-2002, 2002-2006, 2007-2011, 2012-2016, 2017-2020). Computing annual trends in sea level rise is challenging as annual trends are very sensitive to the time-period chosen due to the very high inter-annual variability in global mean sea level, and thus we refrain from calculating annual trends in GMSL.

**Action:** We added in Table 2 the fraction of sea level rise driven by the ice sheets in brackets next to the rates of ice sheet mass change.

**Minor Comments**

lines 64-68: In the abstract you switch between stating mass change for GrIS as a positive mass loss number and for Antarctica as negative numbers to also indicate mass loss. Make sure you are consistent with sign convention throughout.

**Response:** Thank you, we have corrected this in the abstract.

line 99: I prefer the use of the Oxford coma in sentences because I think it makes them easier to read. It is apparently not favored by these authors and I normally accept that stylistic preference, but there are several instances in this paper where the additional coma would help with sentence flow. For example, I had to read this particular sentence a few times. I recommend it is changed to "…geophysical corrections, SMB models, or GIA models in …"

**Response:** We have corrected this sentence and added comas throughout the manuscript.

line 152: Instead of "1 input-output method estimate" you could say "the input-output method estimate"

**Response:** Corrected.

lines 125-144: I appreciate the summary of the methods and their strengths and weaknesses!

**Response:** Thank you!

line 190: How did you quantify linear model structural error?

**Response:** We quantified linear model structural error as the standard error of the regression, we added this in the sentence.

**Action:** This sentence now reads 'The error on the derived time-series is taken as the regression error which incorporates the original measurement error and the linear model structural error computed as the standard error of the linear regression'.

line 223: Why are you only reporting differences from 2007-2011? The previous sentence states the datasets have a much longer period of overlap.

**Response:** The input-output estimates span together the period 1992 to 2020 but only overlap during the period 2007 to 2011 (Mouginot's estimate spans the full period 1992 to 2020, Colgan's the period 1995 to 2020, and Andersen's the period 2007 to 2012). We have clarified the two sentences in the revised manuscript to avoid confusion.

**Action:** We have clarified the sentences as: 'We include 3 input-output method estimates of GrIS mass balance, all at annual resolution and that together span the period 1992 to 2020 and overlap during the period 2007 to 2011. During their common period, annual rates of mass change determined from these three input-output datasets have a median difference of 28.5 Gt yr$^{-1}$ with a standard deviation of 35 Gt yr$^{-1}$.'

Figure 1: While I really like the idea of this table, I struggled to see the aggregate average (black) when there are a large number of gravimetry estimates (green). Consider changing the shade or saturation of the green color. The aggregate average also needs to be stated in the caption and the difference in y-axis scaling should be noted as well. This is a stylist preference but I recommend only plotting the y-axis labels once per side to reduce clutter.

**Response:** We have changed the shade of the green colour, updated the caption, and plotted the y-axis labels once per side as recommended. Thank you for the suggestion.

**Reply to Kenneth Mankoff comments (CC1 and CC2)**

Thank you for your detailed comments and suggestions and for your contribution to the next IMBIE assessment.

We divide the comments posted between comments pertaining to the present preprint and comments pertaining to the framework of the Ice Sheet Mass Balance Inter-Comparison Exercise (IMBIE):

- We welcome comments on this preprint and have revised our manuscript to accommodate those where necessary.
- We also welcome comments pertaining to the IMBIE assessment framework, however we cannot fully resolve those general comments within the scope of this manuscript. This preprint was submitted with the intent to report on the latest IMBIE results, rather than designed for implementing changes to the existing framework at this stage of the process. Altering the IMBIE assessment strategy is a long-term process involving a large number of participants and is outside the scope of this publication and its peer-review process. Any changes to the IMBIE framework, such as making all individual datasets available or changing the IMBIE error budget, should be decided after a consultation process with the IMBIE Executive Committee and participants. However, we do appreciate those comments and have added a new section in the Discussion on the limitations of our dataset and propose a roadmap for future improvements that could be implemented in future IMBIE assessments. We detail in our responses below what changes we can reasonably accommodate at this stage and what changes we deem outside the scope of this publication and how we could address those in future assessments.

I want to start declaring some conflicts of interest I have with this paper, but I have thoughts and comments that I would like to give, so I am providing this public review.

Conflicts: I am the Chief Editor (Ice) for ESSD, but have declined to edit this paper because I contributed to some data used in it - but not enough that I should be a co-author. I am also a contributor to the next version of IMBIE. I also make reference to my own recent paper and suggest citation of it, and correction of some of your text based on it. Please take all this into consideration while reading my review, comments, and suggestions. I am choosing to submit this as a "community comment" not an "Chief editor comment".

**CC1.1** "A to B", in this case "1992 to 2020" is ambiguous. Or at least not as clear as it could be. Is 2020 included or not? I suggest changing all "to" to "through", as in "1992 through 2020".

**Response:** Yes, 2020 is included. We prefer to keep the English formulation 'from 1992 to 2020' rather than the suggested US English, but we have clarified in the introduction of the paper that it does include year 2020.

**Action:** We clarified this in the Introduction: 'Here, we extend these records to cover the same extended period (1st January 1992 to 31st December 2020) for both ice sheets. In the rest of the paper, all of time periods cited refer to the period extending from 1st January of the first year quoted to 31st December of the second year quoted.'

**CC1.2** Last line of abstract should follow ESSD standard: Cite data product.

**Response:** Thank you for pointing this out. We added a citation to our data product in the abstract.

**Action:** We added the following sentence in the abstract: 'The dataset is publicly available at https://doi.org/10.5285/77B64C55-7166-4A06-9DEF-2E400398E452 (The IMBIE Team, 2021).'

**CC1.3** L101-103: Again, "through" rather than "to" so it is clear the last year is included.

**Response:** Please see our previous comment.

**CC1.4** L135: When discussing IO method you should probably cite Mankoff /et al./ (2021). L137 mentions "year-to-year" but Mankoff /et al./ (2021) show that IO can provide daily estimates of mass change. L138 " The technique provides moderate (annual) temporal sampling" <-- Or daily, or whatever resolution the RCMs output. I do take 12-day velocity data and resample to daily, which you may have issue with. But even 12 day is more frequent than annual.

**Response:** Agreed. There has been considerable progress in improving the temporal resolution of the IO method recently. Most estimates are now provided at monthly resolution and Mankoff et al. (2021) even provides daily estimates of Greenland's mass balance (which we cite in our revised manuscript). The temporal resolution of the IO method is dependent on the temporal resolution of the ice discharge estimate, which is updated depending on the revisit time of the satellite passes (e.g. 6-days when using Sentinel 1a and 1b or 12 days when using Sentinel 1a only), and on the temporal resolution of the surface mass balance model estimate used. For achieving a daily resolution, resampling the satellite velocity data is necessary as mentioned in the comment, and therefore we prefer to characterise the temporal sampling of the IO method as monthly, based on the temporal resolution of the input datasets rather than based on the temporal resolution of resampled datasets. We have revised the corresponding text to discuss the temporal resolution of the IO method.

**Action:** We added the following text to discuss the temporal resolution of the IO method and highlight the progress made in this regard in recent years: 'During the last decade, new satellite missions with a more frequent revisit time (up to 6 days using image pairs from Sentinel-1a and Sentinel-1b available during the period 2016 to 2021 until the end of Sentinel-1b mission) have been used to improve the temporal resolution of ice velocity measurements, allowing for the investigation of seasonal fluctuations in ice velocity (King et al., 2018; Lemos et al., 2018) and produce of monthly estimates of ice discharge at the continental scale. Mankoff et al. (2021) even produced daily estimates of ice sheet mass balance from the input-output method by resampling the velocity data, however the original temporal resolution of ice velocity measurements does not exceed 12 days.'

**CC1.5** Paragraph 1 of section 2 is methods or background, not data.

**Response:** This paragraph of background text helps the reader understand the nature of the datasets we are aggregating in this study (altimetry, gravimetry, input-output method), instead of jumping straight into higher-level details of the different datasets included. We prefer not to move this paragraph later in the text to the method section, as this section is dedicated to the methods employed for aggregating the different datasets. However, we have now divided the data section into 'Input Data' and 'Output Data' as suggested in another comment to improve the overall clarity of this section.

**CC1.6** Paragraph 2 of section 2 is intro to data. It would be good to talk about the actual input data. Your Appendix Table A1 is appropriate as an Appendix in other journals, but is the core of an ESSD product, and should not be hidden in an Appendix. This should be in Section 2, "Data".

**Response:** Agreed, thank you for the suggestion.

**Action:** We have moved Table A1 to Section 2.1 Input Data rather than placing in the Appendix.

**CC1.7** Feel free to split "Data" into "Input Data" and "Output Data".

**Response:** Thank you for this suggestion, we have revised the manuscript accordingly.

**Action:** We split the 'Data' section into 'Input Data' and 'Output Data'.

**CC1.8** Paragraph 3 of Data talks about masks and ROIs. Can you share these? I think not, because each data set used their own and then told you the area of the basins, but did not provide you with the boundaries themselves (is this correct?). But it may be worth pointing out that many different input products may have used many different masks.

**Response:** We do provide on our website the ice sheet masks to use at http://imbie.org/imbie-3/drainage-basins/ and all participants are asked to use either the Zwally (2012) or Rignot (2011a, 2011b) ice sheet delineations. Previous IMBIE assessments have shown that using either the Zwally or Rignot masks leads to a difference less than 1.1 % in mass balance, see L164-L166 in our preprint.

**CC1.9** No mention of peripheral glaciers, and their inclusion or exclusion from each of the 50 products. Does this explain some of the disagreements?

**Response:** We have added a discussion of the peripheral glaciers and ice caps in our Discussion section. All gravimetry estimates include the peripheral glaciers and ice caps as the coarser spatial resolution of GRACE is not sufficient to distinguish between the ice sheet and the nearby glaciers and ice caps. On the other hand, the altimetry estimates do not include those and finally some of the input-output estimates include them. We added some text to reflect on that and on the impact of the inclusion/exclusion of the peripheral glaciers on our estimate and on the agreement/disagreement between techniques.

**Action:** We added a paragraph discussing the inclusion/exclusion of the peripheral glaciers and ice caps and outlined future approach to solve this issue (as also suggested by by Romain Hugonnet and Etienne Berthier):
'In this section, we discuss the limitations of our dataset and a roadmap to improve ice sheet mass balance assessments. The inclusion of the peripheral glaciers and ice caps in the vicinity of the Greenland and Antarctic Ice Sheets is ambiguous in our assessment as not all individual estimates of ice sheet mass balance included here account for those. This relates to the varying ability of satellite techniques to resolve mass balance over those small glaciated areas. Space gravimetry has a coarse spatial resolution of a few hundred kilometres which is not sufficient to separate signals of mass change originating from the ice sheet and its peripheral glaciers. On the other hand, the altimetry estimates included in this assessment exclude the peripheral glaciers and ice caps due to the complex terrain of these glaciers and their relatively small size compared to the footprint size of traditional pulse-limited altimeters. Finally, the input-output estimates do include mass changes from these glaciers, mostly by estimating their changes in SMB. Despite covering a relatively small area (around one tenth of the area of the ice sheets) (Pfeffer et al., 2014), these glaciers contribute significantly to global mean sea level rise with ice losses originating from the Greenland and Antarctic Ice Sheets amounting to $36 \pm 6$ Gt yr$^{-1}$ and $21 \pm 5$ Gt yr$^{-1}$ during the period 2010-2019, respectively (Hugonnet et al., 2021). In addition, ice losses have accelerated in the periphery of the Greenland Ice Sheet, with glacier mass loss increasing by 64 % between 2003-2009 and 2018-2021 (Khan et al., 2022). These glaciers therefore need to be accounted for without ambiguity in future IMBIE assessments to remove systematic biases between the different satellite techniques linked to their (non-)inclusion in individual mass balance estimates. Recent progress in satellite altimetry, with the development of CryoSat-2 swath radar altimetry for measuring mass changes of mountain glaciers (Foresta et al., 2016; Jakob et al., 2021) and the launch of ICESat-2, already contribute to a better mapping of those glaciers. New community initiatives, such as GlamBIE (the Glacier mass balance Inter-comparison Exercise), will further contribute to separating mass changes between the ice sheets and glaciers lying at their periphery by offering a consensus-estimate that could be removed from the gravimetry estimates that currently account for both.'

**CC1.10** I believe RACMO has a binary ice sheet mask: 1 or 0. On the other hand, MAR has a floating point mask, and it is up to MAR users to decide if the cutoff for "ice sheet" is 0.5 or some other value. Is this worth discussing? Does this explain some of the differences between estimates?

**Response:** This is outside the scope of the present manuscript. The ice sheet mask used in the different SMB models should in principle not impact on our comparisons as we require all participants to submit their ice sheet mass balance estimates using the Rignot or Zwally basins. However, it is true that in the IO method, the SMB component which is estimated using a SMB model (MAR, RACMO or another SMB model) could differ due to different ice sheet masks used. It is hard to disentangle whether the use of a different ice sheet mask has a significant impact on the different IO estimates included as other factors such as the positions of the flux gates, the thickness dataset used, the interpolation/scaling scheme used, the reference time-period chosen to calculate the SMB anomaly, are likely to have a larger impact than the ice sheet mask used on differences between IO estimates. We are currently working on a SMB model inter-comparison for the next IMBIE assessment and are planning to define a common ice sheet mask to use when submitting SMB model estimates to future IMBIE assessments.

**CC1.11** Table 1: "X" and gray is redundant. Could be visually cleaner if you just did gray and no "X"?

**Response:** Agreed

**Action:** We have amended Table 1 and remove the 'X'.

**CC1.12** 3 Methods: I am happy to see that you shared your code. Maybe mention this here, and even reference specific functions in the code? Code should not just be on GitHub, where it is likely to change. Or if it is, reference a specific git hash. Or export from GitHub and release a 'frozen' version on Zenodo or some other service where you can DOI your code.

**Response:** Agreed.

**Action:** Following your suggestion, we have exported a frozen version of our core on Zenodo and provides the DOI in the Methods section: 'The IMBIE assessment software used to produce the dataset presented in this study is available at https://doi.org/10.5281/zenodo.7342481.'

**CC1.13** IMBIE has the opportunity here to do really transformative "open science" and set a standard for how it could be done. Can you ask all 50 data providers if they are willing to share (publish) the data that they provided with you? If so, you could provide the input data, and the full processing pipeline to generate the output data. This would let people re-run the analysis but with different methods and assumptions, if they choose.

**Response:** IMBIE is certainly an "open science" activity, and our primary objective is to make a single community assessment of ice mass balance available to the wider scientific community. Unfortunately, to share all of our constituent datasets we would need to amend the rules of participation which, in turn, would require consultation and approval of all IMBIE participants. This is not possible at this stage, but we will consider this option for future assessments. Nevertheless, all of the individual products aggregated in our assessment have necessarily been published and should therefore be available from the authors. We recognise that this is something we could facilitate and will recommend that sharing of constituent data become a factor of future assessments.

**CC1.14** L196/197: " The associated error is calculated as the root mean square of the contributing time-series errors." I take this to mean that errors reduce in quadrature? And that as you add more data products, your errors decrease? I am not sure that assuming all errors are random, and that more measurements reduces error, is reasonable. It is quite likely that there are some biases in the data, that remain with the same sign through time, or are the same for different products.

**Response:** Yes, in our error budget, the errors decrease when adding more data products: if several independent products give a similar mass balance estimation, we assume that we can have more confidence in this estimate. We agree that our assumption that all errors are random is simplistic and that our error budget could be improved, but this is outside the scope of this paper which follows our most recent approach (The IMBIE Team, 2020). Indeed, all contributors define and compute errors differently and therefore using these individual errors estimates to produce an aggregated estimate is complex and this why we use a simple approach. We are planning on setting up a working group to discuss the IMBIE error budget for future assessments, but this will involve a consultation with the IMBIE consortium, which will require time and effort to design and implement a new error budget and is therefore outside the scope of this paper. By comparing datasets generated using different techniques (altimetry, gravimetry, input-output method), we are trying to identify biases in the data but, as each estimate uses different corrections/auxiliary datasets, identifying biases in the data is not straightforward.

**Action:** We outline a roadmap on how to further reconcile ice sheet mass balance estimates in future IMBIE assessments in Section 5.3, including improving the error budget of individual ice sheet mass balance estimates and of the reconciled IMBIE estimate: 'Producing estimates with a better temporal resolution by using data from the newest satellite missions, reprocessing the satellite record with the newest geophysical corrections, and using a better uncertainty characterisation, will undoubtedly help further reconcile satellite assessments of ice sheet mass balance produced from different techniques.'

L202: See previous comment.

**Response:** See previous comment.

**CC1.15** Fig 1: Can remove all but one Y axis labels (L & R) since they are all the same.

**Response:** Agreed, we kept the Y-axis on the top panel only.

**Action:** We modified the figure accordingly with only one Y axis labels.

**CC1.16** Fig 1: 2020 shows 1 method, but the bar is 'black' implying 'all'. Does this mean "all" is not "all methods" but "average" or "mean" or "median" of "all available data in a given year"? Or something else?

**Response:** Here 'All' refers to the final reconciled estimate produced by combining all the datasets together, we have renamed the black line as 'This study' and updated the caption.

**Action:** We changed the label on the Figure and the caption now reads: 'Annual rates of mass change of the (a) GrIS, (b) AIS, (c) APIS, (d) EAIS, and (e) WAIS from the altimetry, gravimetry and input-output estimates included in this study (shown by the coloured bars) and the reconciled estimate produced from combining those estimates (shown by the thick black bars). The estimated 1$\sigma$, 2$\sigma$, and 3$\sigma$ ranges of our final reconciled estimate are shaded in dark, mid and light grey, respectively. The number of individual mass balance estimates collated at each epoch is shown below each bar.'

**CC1.17** Section 5 Discussion Paragraph 2 and Figure 4: I'm not sure this is relevant or appropriate for ESSD - It is science outside of the dataset. I would reframe Section 5 Paragraph 1 as "Validation" - basically admitting you cannot easily validate this against anything because you've incorporated all datasets, or if you did validate against the one not incorporated (Mankoff /et al./, 2021) it would only be useful in pointing out issues with that dataset, not the 50 that make up your dataset. Perhaps the last paragraph of Section 4 could be combined with this - there you basically validate against the last version of IMBIE. I'm not sure this is a "Result".

**Response:**

On the validation of our estimate:
- As pointed out by this comment, we cannot validate our reconciled mass balance estimate as we are trying to include all mass balance estimates available at the time of the start of this exercise. Therefore, it does not make sense to compare this reconciled estimate, which combines 50 independent estimates, against a single estimate. However, the individual datasets aggregated in this manuscript have all been published in peer-reviewed publications and thus for the vast majority of them, some kind of validation has already been performed as part of these individual studies.
- We do compare our estimate against the previous version of IMBIE to assess the impact of changes made to the processing scheme and document how our previous and current estimates differ. While this is not a validation per se, we agree that it would fit better in a section separate from the main 'Results'. We have moved this section to the Discussion section.

On the relevance of Section 5, Paragraph 2: In this paragraph, we compare our new observational estimates to the IPCC projections of future sea level rise. This illustrates the purpose of our study – providing a reconciled estimate of ice sheet mass balance derived from satellite observations that can easily be used by the ice sheet modelling community and sea level scientists to compare their simulations against contemporary observations without having to choose between the numerous products available. This comparison also provides useful information for future studies in terms of both observational and modelling studies. We believe that this paragraph does belong to the 'Discussion' section of this paper as we are simply comparing the estimate presented in the manuscript to the latest IPCC projections and reflecting on the synergies between observations and projections, rather than doing 'new science'. We leave it to the editor to decide on whether it adds value.

**Action:** We have reorganised Section into three subsections to improve the clarity and flow of the paper:
- **5.1 Comparison to previous IMBIE assessments**
- **5.2 Comparisons to sea level contribution and projections of future sea level rise from this new dataset**
- **5.3 Limitations of this study and roadmap for future improvements,** in which we address the potential sources of disagreement between the three techniques (e.g. (non-)inclusion of the peripheral glaciers and ice caps, missing ice thickness measurements for the IO method, GIA and SMB corrections, error budget of the aggregated estimate) and propose a roadmap to address these issues in future IMBIE assessments

**CC1.18** L398 Acknowledgements: This should probably be more comprehensive given the length of your author list.

**Response:** Agreed.

**Action:** We have added a more comprehensive acknowledgments section.

**CC1.19** Figure A1: This highlights what I believe is a significant deficiency in your error handling. It appears that when you have fewer products, your errors decrease. Shouldn't your uncertainty increase when you're relying on only 1 product?

**Response:** The caption of Figure A1 (now Figure 1 in our revised manuscript) was incorrect, many thanks for spotting this! This figure shows the aggregated mass change rate series for each individual estimate included in this study and the grey shadings represent the 1-sigma, 2-sigma, 3-sigma uncertainty of the aggregated time-series and not of the final reconciled estimate (which incorporates the three techniques altogether) as previously stated.

We calculate errors on the three technique dependent time-series (resulting from aggregating all altimetry estimates together, all gravimetry estimates together, all input-output estimates together) differently than how we calculate errors on the final reconciled time-series (the combination of the three technique dependent time-series):

- To calculate the errors on each technique-dependent time-series, we compute the root-mean-square of the contributing time-series errors. Thus our uncertainty decreases when the number of datasets increases (as we divide by $\sqrt{n}$, with n the number of datasets at the given epoch). For instance, for the Greenland IOM aggregated time-series, when we only have one dataset we simply take the input error as the error for the aggregated time-series (e.g. at the very beginning of the time-series when we have only the Mouginot estimate, the error is the error from Mouginot). When we have multiple datasets, we sum their errors in quadrature and divide by $\sqrt{n}$ so the resulting error is dependent on both the input errors from the individual datasets and the number of estimates available at that epoch.

- The error on our reconciled estimate (produced by reconciling the altimetry, gravimetry and input-output aggregated time-series) decreases according to the number of independent estimates, as we divide our error at each epoch by the root-mean-square of the number of independent techniques available (i.e. from 1 to 3). For year 2020, when we only have one gravimetry estimate, we simply take the input error on that particular estimate. Therefore, it is important to note that our errors are also highly dependent on the errors of the input datasets that we incorporate, and not only on the number of datasets included.

**Action:** We corrected the corresponding caption: 'Individual rates of ice sheet mass balance from the input-output, altimetry, and gravimetry groups over the GrIS, APIS, EAIS, and WAIS included in this study. The grey shading shows the estimated 1σ, 2σ, and 3σ ranges of the aggregated time-series per group in dark, mid, and light grey, respectively. The uncertainty is calculated as the root mean square of the contributing errors at each monthly epoch.'

In addition, the three technique-aggregated time-series are now also shown separately with their respective uncertainties in the Appendix (following comment RC1.1 by Anny Cazenave).

**CC2.1** One additional comment: ISO 8601 is a really nice date standard. Is there a reason for using yyyy.dec rather than yyyy-mm-dd?

**Response**: We define regular epochs of 1/12 years, which is why we prefer to use yyyy.dec rather than yyyy-mm-dd.

**Response to Romain Hugonnet and Etienne Berthier's comment (CC3)**

We commend the authors for continuing to develop the IMBIE effort and provide a multi-technique estimate of ice sheet mass balances. We leave it to the reviewers to evaluate the study in detail.

Thank you for commenting on our manuscript and bringing up the topic of the peripheral glaciers.

**CC3.1** The authors explain how the two ice sheets were split into different basins (i.e. two sets of ice sheet drainage basins were used). However, as in earlier IMBIE studies, *they did not explain how they took into account (or not) the mass changes of glaciers peripheral to the ice sheets* (Rastner et al., 2012; Pfeffer et al., 2014; Gardner et al., 2013). This issue is important because the three techniques have different spatial resolution and hence varying capabilities to separate the mass changes from the main ice sheets and the glaciers lying at their periphery. Our understanding is that gravimetric studies include peripheral glaciers, altimetric studies exclude peripheral glaciers, and input–output studies do both. Therefore, there might be important **systematic errors** in the IMBIE estimates.

This is relevant for both ice sheets, but especially for the Greenland Ice Sheet where the losses from peripheral glaciers amounted to 36 ± 6 Gt/yr (95% confidence) during 2000–2019 (Hugonnet et al., 2021). This was independently assessed at 27 ± 12 Gt/yr during 2003–2010 and 42 ± 12 Gt/yr during 2019–2022 (Khan et al., 2022). A loss of 36 Gt/yr translates to about **19%** of the overall Greenland Ice Sheet mass loss over the period of 2000–2019, and is more than twice the uncertainty range of ± 16 Gt/yr provided by Otasaka et al. for 1992–2020. We foresee that removing the mass contribution of peripheral glaciers (in particular for gravimetry-based estimates of Greenland and the Antarctic Peninsula) will increase the uncertainties.

To conclude, the authors should provide a clear definition of the Greenland and Antarctic ice masses for which they estimate mass losses for each of the applied techniques. This would avoid double counting the mass change from peripheral glaciers when IMBIE results are combined with glacier-specific mass change estimates to evaluate closure of the sea level budget.

**Response:** Yes, you are right that not all ice sheet mass balance estimates included in this study account for the peripheral glaciers and ice caps. The gravimetry estimates include both, as the spatial resolution of GRACE/-FO is too coarse to distinguish between the ice sheet and its peripheral glaciers and ice caps. The altimetry estimates exclude the peripheral glaciers. Finally, some of the input-output estimates do include the peripheral glaciers of Greenland but not all; and in Antarctica, the only input-output estimate that we include in our assessment accounts for both the ice sheet and the peripheral glaciers.

Dividing Greenland and Antarctica into 'ice sheet' and 'peripheral glaciers and ice caps' is an ongoing debate within the glaciological community; this distinction mainly exists due to the challenges associated with observing these very small glaciers in comparison to the vast ice sheets. However, recent progress in satellite altimetry (the development of CryoSat-2 swath altimetry and the launch of ICESat-2) has already led to improve mapping of those small glaciated areas. In addition, we are also hoping to make use of the outputs from the GlamBIE (Glacier mass balance Inter-comparison Exercise) project, which will produce a reconciled estimate of the peripheral glaciers of Greenland and Antarctica. We have added a paragraph in our Discussion section in which we discuss the (non)-inclusion of those glaciers in our assessment (see Section 5.3 of our revised manuscript) and this can be improved in future IMBIE assessments.

**Action:** We added a paragraph in the Discussion section as also suggested by Ken Mankoff. Please see our response to their comment for our full response (please see our response to CC1.9)

References

Gardner, A. S., et al.: A Reconciled Estimate of Glacier Contributions to Sea Level Rise: 2003 to 2009, Science, 340, 852–857, https://doi.org/10.1126/science.1234532, 2013.

Hugonnet, R., et al.: Accelerated global glacier mass loss in the early twenty-first century, Nature, 592, 726–731, https://doi.org/10.1038/s41586-021-03436-z, 2021.

Khan, S. A., et al.: Accelerating Ice Loss From Peripheral Glaciers in North Greenland, Geophysical Research Letters, 49, e2022GL098915, https://doi.org/10.1029/2022GL098915, 2022.

Pfeffer, W. T., et al. : The Randolph Glacier Inventory: a globally complete inventory of glaciers, J. Glaciol., 60, 537–552, https://doi.org/10.3189/2014JoG13J176, 2014.

Rastner, P., et al.: The first complete inventory of the local glaciers and ice caps on Greenland, The Cryosphere, 6, 1483–1495, https://doi.org/10.5194/tc-6-1483-2012, 2012.

---

## Author Response (AR2)

We would like to thank the editor for his helpful comments, which have improved the clarity of our paper. Please find below our responses in blue and the changes we made in green below the comments.

**Major comments**

I would like to thank the reviewers and the community members for providing thoughtful and constructive comments. The authors have done a lot of work to address these comments in the revised manuscript and I think that this has improved the paper. However, some significant concerns were raised about the treatment of uncertainty and the potential for large systematic errors in the aggregated mass balance results, and your response to reviewers does not appear to sufficiently address these concerns. Before this paper can proceed to final publication several points need to be addressed.

First, the methods section is quite brief and does not fully and clearly describe the procedures used. This should be expanded and the treatment of uncertainty, strengths and limitations, and rationale for the approach should be directly addressed here rather than left for later in the discussion. Also, other parts of the paper are quite detailed in describing and interpreting the results and begin to go beyond the scope of ESSD. Specifically, "Articles in the data section may pertain to the planning, instrumentation, and execution of experiments or collection of data. Any interpretation of data is outside the scope of regular articles. Articles on methods describe nontrivial statistical and other methods employed (e.g. to filter, normalize, or convert raw data to primary published data) as well as nontrivial instrumentation or operational methods. Any comparison to other methods is beyond the scope of regular articles." By expanding the methods description and cutting some of the discussion, this would better address reviewer concerns, focus the paper more directly on how the aggregated data product was derived, and put the treatment of uncertainty upfront.

- In our revised manuscript we have explained our treatment of uncertainty in more detail, and we have assessed the potential for systematic errors associated with mass changes from peripheral glaciers and ice caps. Our method section (Section 3) is now expanded to contain a more detailed description of our estimated errors at each step. Peripheral ice masses have a small (<5 %) impact on our mass balance assessment because their signal is only a potential omission in gravimetry-based estimates (i.e. one third of our reconciled estimate), and because the mass change owing to glaciers and ice caps is in any case small relative to the mass change of the ice sheets. The potential bias is within the uncertainty bounds of our reconciled estimate. While our participants make efforts to account for the mass loss, it remains challenging to do so for those based on satellite gravimetry, and this remains an area of future research.
- We have removed the comparison of the results to projected sea level rise and associated figure and table.

**Specific Comments**

**Figure 1:** The caption says that uncertainty is calculated as the RMSE at each monthly epoch while the shading represents the standard deviation of the aggregated time series. It isn't clear what the measure of uncertainty is to me. In Figure 4 the uncertainty is quantified as the standard deviation. Or is sigma not standard deviation? Later in the text it is defined as that (P16, L326). This should be

better explained in the text and captions so that it is more clear what is shown. This was also raised in CC1.19, but the action taken didn't quite clear this up in the paper.

**Response:** We agree that the caption is not very clear. Here (and on Figure 4 as well), sigma is not the standard deviation but the measure of our uncertainty. On Figure 1, the uncertainty plotted is the uncertainty of our aggregated time-series per satellite technique as calculated in step (ii) of our methods, which is calculated as sum in quadrature of the contributing individual time-series errors divided by the square root of the number of estimates. We have clarified our uncertainty calculation in the text, detailing the calculation at each step and have clarified the caption of Figure 1. We also removed from Figure 1 the different grey shadings and kept only the darkest grey shading representing the uncertainty to improve the clarity of the figure.

**Actions:**
- We have clarified the caption of Figure 1 as:

Individual rates of ice sheet mass balance from the input-output, altimetry, and gravimetry groups over the GrIS, APIS, EAIS, and WAIS included in this study and standardised following the procedure described in Section 3 (i). The grey shading shows the estimated uncertainty of the aggregated time-series per group calculated following the procedure described in Section 3 (ii).

- We removed the different grey shadings and kept only the shading representing the uncertainty of the aggregated group estimate.

In regards to **CC1.5**, it might be useful to add a sub-heading under data called background for the first paragraph.

**Action:** We added this sub-heading.

In regards to **CC1.8**, the link you provided in your response could be added to the text of the manuscript.

**Action:** We added the link to the drainage basins description in the text (section 2.2 Input data).

**Section 3 (Methods):** This is a fundamental section of the paper and is essential for understanding this data product and its quality. The error analyses should be better defined, clearly described, and some rationale or justification should be provided.

**Response:** We have revised this section and added details in each sub-section on our error calculation and added a new paragraph describing how we account for the inclusion of the peripheral glaciers and ice caps in our error budget. Below we detail the changes made in each sub-section.

**3 i)** Computing time-series of mass trends: Is the output of this step what we see in Figure 1? There is some error computed for each series—is this shown anywhere? Is the uncertainty (sigma) that is shown in Figure 1 an output from step ii? I am not clear how the error incorporates the original measurement error.

**Response:** Figure 1 shows the dM(t)/dt time-series derived from step (i), but the errors plotted on top are not the individual error time-series but the aggregated errors calculated from step (ii). This choice stems from the fact that if we were to plot the individual error time-series calculated from step (i),

there would be too many overlapping shading, especially for the gravimetry group, and the figure would not be easy to read. We plotted the figure with individual errors for reference below:

[Figure]

Regarding the uncertainty calculation for step (i): at each epoch, the error on the derived dM(t)/dt is computed as the sum in quadrature of the standard error of the linear regression and the mean of the input errors falling with the 36-month sliding window. This second term thus incorporates the original measurement error.

**Actions:**
- We have clarified the caption of Figure 1 (see also our response to your previous comment on Figure 1)
- We have revised the text describing the error calculation of step (i) as:

'The error on the derived time-series is taken as the sum in quadrature of the linear model structural error computed as the standard error of the linear regression $s_e$ and the mean of the errors of the $n_w$ points in the original ΔM(t) time-series falling within the 36-month sliding window as:

$$\sigma_{\frac{dM}{dt}}(t) = \sqrt{s_e{}^2 + \left(\frac{1}{n_w}\sum_{i=0}^{n_w-1}\sigma_{\Delta M,i}\right)^2} \tag{1}$$

**3 ii)** Aggregating time-series of mass trends from similar satellite observations: This section could be expanded and made more clear. Please explain how the error-weighted averages were computed. It says that this was done using the same approach—as in step i? That was a linear trend fitting approach as I understand. Then the overall error of aggregate series is RMSE of contributing time series. Is this the uncertainty we see in Figure A1?

**Response:** This is a misunderstanding from our text not being clear enough. The original sentence was: 'We calculate each aggregated time-series by taking the error-weighted average of monthly rates of ice sheet mass change computed using the same technique'. Here, 'the same technique' referred to the altimetry, gravimetry, or input-output method, and was not meant to be read as 'the same approach as step (i)'. We removed this to improve the clarity of the section and detailed further our uncertainty calculation. The uncertainty calculated in step (ii) is the uncertainty shown in Figure A1 (and in Figure 1). We clarified further our description of the errors calculation: we define the errors on the aggregated time-series as the sum in quadrature of the errors of the contributing individual estimates divided by the square root of the number of estimates used in the aggregated product, rather than describing it as the root-mean-square of errors, which could be mistaken as the RMSE.

**Actions:**
- We corrected the corresponding text and added the equations used for deriving the aggregated time-series for each satellite technique and the corresponding error:

We aggregate the standardised time-series of mass trends within the altimetry, gravimetry, and input-output groups separately to produce three time-series over each ice sheet region $\frac{dM_{aggr}(t)}{dt}\Big|_{group}$, where $group$ refers to one of the three independent satellite techniques (i.e. altimetry, gravimetry, or input-output method). We calculate each aggregated time-series by taking the error-weighted average of the $n_{estimates\ per\ group}$ individual monthly rates of ice sheet mass change available from the same satellite technique group at each month:

$$\frac{dM_{aggr}(t)}{dt}\Big|_{group} = \frac{\sum_{i=0}^{n_{estimates\ per\ group}-1} \frac{dM(t)}{dt}\Big|_{group,i} \Big/ \sigma_{\frac{dM(t)}{dt}\big|_{group,i}}}{\sum_{i=0}^{n_{estimates\ per\ group}-1} 1 \Big/ \sigma_{\frac{dM(t)}{dt}\big|_{group,i}}} \qquad (2)$$

The associated error is calculated as the sum in quadrature of the contributing individual time-series errors belonging to the same group divided by the square root of the number of estimates in the group:

$$\sigma_{aggr,group}(t) = \sqrt{\frac{1}{n_{estimates\ per\ group}} \sum_{i=0}^{n_{estimates\ per\ group}-1} \sigma^2_{\frac{dM(t)}{dt}\big|_{group,i}}} \qquad (3)$$

- We changed the caption of Figure A1 to:

Mass balance time-series from the aggregated altimetry, gravimetry and input-output method over the a) WAIS, b) EAIS, c) APIS, and d) GrIS. The vertical dashed lines mark the overlap period of the three time-series. The aggregated time-series and corresponding uncertainties are calculated following the methods described in Section 3 (ii).

**3 iii)** Combining the altimetry, gravimetry, and input-output time-series of mass trends: Here again it is unclear how the error-weighted mean was calculated. Then error of the reconciled mass series is estimated as RMSE divided by number of techniques. What is the rationale or justification for this? Or this arbitrary? Then rate of annual mass balance and that over certain epochs is calculated and error is the average of contributing error divided by square root of the number of years of the time period. Specifically, why? It isn't clear and by this point it is difficult to track how errors have been accumulating (or been neglected) throughout the process. Finally the mass trends are summed over multiple ice sheets and the error is the root sum square of uncertainties for each region. What about the concerns raised by the reviewers about peripheral ice masses and their change?

**Response:** We feel it is important to stress that the methods we employ are well established in the literature (see e.g. Rignot et al. (2019), IPCC AR5 (Vaughan et al., 2013) and AR6 (Fox-Kemper et al., 2021), and previous IMBIE assessments (Shepherd et al., 2012; The IMBIE Team, 2018, 2020)). Furthermore, because we can only make significant changes to our core methods by completing a formal consultation with our project participants which we accommodate in annual cycles, we cannot change significantly the methods we employ to compute uncertainties. However, we have clarified our uncertainty calculation and we now discuss the impact of the peripheral ice masses in our assessment.

We have added the formula used to calculate the error-weighted mean and we have clarified the description of our error calculation on the reconciled altimetry, gravimetry, and input-output mass trends time-series: we compute the error on the reconciled mass trend time-series at each epoch as the sum in quadrature of the aggregated time-series errors divided by the square root of the number of independent estimates available. We added the equation in each sub-section of Section 3, which hopefully improves the clarity of our error characterisation and makes it easier to track how errors have been propagated throughout.

Regarding the peripheral ice masses and their change, we added a new paragraph at the end of Section 3 to discuss the impact of their inclusion in gravimetry estimates. We use the estimate from Hugonnet et al. (2021) to remove the contribution of the peripheral glaciers and ice caps on our aggregated gravimetry time-series and re-combine this modified gravimetry time-series with the altimetry and input-output aggregated time-series. We find that removing the peripheral ice masses has a small impact on our final reconciled mass balance estimates with a reduction in the rate of mass loss of less than 10 Gt yr$^{-1}$ in Greenland and less than 3 Gt yr$^{-1}$ in Antarctica, which is smaller than our uncertainty estimate.

**Actions:**
- We clarified our error calculation for the reconciled mass trends time-series:

We combine the altimetry, gravimetry, and input-output time-series to produce a single reconciled time-series of mass trends by taking the error-weighted mean of the $n_{group}$ independent estimates for which a mass trend estimate is available at each epoch (comprised between 1 and 3):

$$\frac{dM_{reconciled}(t)}{dt} = \frac{\sum_{i=0}^{n_{group}-1} \frac{dM_{aggr,i}(t)}{dt} / \sigma_{aggr,i}(t)}{\sum_{i=0}^{n_{group}-1} 1/\sigma_{aggr,i}(t)} \tag{4}$$

We estimate the error on the reconciled mass trend time-series at each epoch as the sum in quadrature of the aggregated time-series errors divided by the square root of the number of independent estimates available:

$$\sigma_{reconciled}(t) = \sqrt{\frac{1}{n_{group}} \sum_{i=0}^{n_{group}-1} \sigma^2_{aggr,i}(t)} \qquad (5)$$

Finally, when summing mass trends of multiple ice sheets, the combined uncertainty is estimated as the root sum square of the uncertainties for each region:

$$\sigma_{total}(t) = \sqrt{\sum_{regions} \sigma_{reconciled,i}{}^2(t)} \qquad (6)$$

- We added a new paragraph in Section 3 in which we discuss the inclusion of the peripheral glaciers and ice caps and its impact on our reconciled assessment:

Here, we discuss the potential systematic bias introduced by the inclusion of the peripheral glaciers and ice caps (GICs) in the gravimetry estimates included in our assessment as the spatial resolution of satellite gravimetry is not sufficient to resolve separately the mass change signals of these two neighbouring ice masses. To examine this further, we use Hugonnet et al. (2021) dataset (https://doi.org/10.6096/13, last access: 23 February 2023) , which provides mass balance estimates of the glaciers located at the periphery of the ice sheets derived from high resolution digital elevation models. During the overlap of Hugonnet et al. study and the gravimetry recorded employed in this study (2002-2019), Greenland peripheral glaciers lost mass at a rate of 35.5 ± 1.6 Gt yr$^{-1}$. In Antarctica (excluding the Sub Antarctic glaciers located further than 1000 km from the ice sheet), peripheral glaciers lost mass at a rate of  11.8 ± 3.4 Gt yr$^{-1}$, 0.7 ± 1.1 Gt yr$^{-1}$, and 5.7 ± 2.5 Gt yr$^{-1}$ at the APIS, EAIS, and WAIS, respectively. To test the impact of the inclusion of the peripheral glaciers in our gravimetry estimates on our reconciled ice sheet mass balance assessment, we use the peripheral glaciers mass trends time-series from Hugonnet et al. to remove the contribution of the GICs on our aggregated gravimetry time-series. We use consecutive 5-year rates of mass change for this analysis and their corresponding uncertainties. For 2020, which is not covered by Hugonnet et al., we use the rate of mass change estimated over the 5-year period 2015-2019 instead. We combine in quadrature the uncertainty on the peripheral GICs mass balance and the uncertainty of our aggregated gravimetry mass balance calculated from Eq. 3. Next, we follow the procedure described in step (iii) to re-combine this modified gravimetry aggregated time-series with the altimetry and input-output aggregated time-series. We compare this modified reconciled estimate to our original estimate and find that removing the contribution of the GICs from the gravimetry time-series results in a reduction in mass loss of 4.1 % and 3.3 % in Greenland and Antarctica, respectively, smaller than the uncertainty bounds of our reconciled estimate (Table A2). This simple analysis shows that the inclusion of the peripheral ice masses in the gravimetry estimates included in this study has a negligible impact on our reconciled mass balance assessment of the WAIS and EAIS, and only a small impact (less than 10 Gt yr$^{-1}$) on our assessment of the GrIS and APIS.

- The rates of mass change computed from both our original reconciled estimate and this modified estimate are presented in a new supplementary table (Table A2):

| Table A2. Rates of mass change (in Gt yr$^{-1}$) over the gravimetry record (2002 to 2020) from our reconciled estimate and from a modified version of our reconciled estimate in which the contribution of the peripheral glaciers has been removed from the gravimetry estimates following the method described in Section 3. | | |
| --- | --- | --- |
| | **Reconciled assessment** | **Modified reconciled assessment** |

| GrIS | -235.6 ± 20.6 | -226.0 ± 20.6 |
|------|---------------|---------------|
| APIS | -18.3 ± 6.0 | -15.7 ± 5.8 |
| EAIS | 6.1 ± 19.7 | 6.2 ± 19.6 |
| WAIS | -104.8 ± 11.2 | -103.6 ± 10.8 |
| AIS | -117.0 ± 23.5 | -113.1 ± 23.2 |

- We modified the first paragraph of our roadmap to further discuss this:

These glaciers therefore need to be accounted for without ambiguity in future IMBIE assessments to remove systematic biases between the different satellite techniques linked to their (non-)inclusion in individual mass balance estimates. Here, we performed a simple analysis to assess the potential impact of the ambiguous inclusion of these peripheral ice masses in our reconciled mass balance assessment and showed that this impact is limited thanks to the fact that we are aggregating different satellite techniques together – including some able to resolve separately ice sheet mass changes – and a different weighting has been applied to the different estimates included. However, future approaches to address this issue will require careful treatment of the leakage of mass signals between the ice sheets and their peripheral GICs within the gravimetry community, rather than being limited to a subsequent removal of the contribution of these glaciers as we have done here. This will nonetheless require robust mass balance estimates for developing and evaluating new methods. The recent inventory of Earth's glaciers from satellite photogrammetry (Hugonnet et al., 2021), recent progress in satellite altimetry – with the development of CryoSat-2 swath radar altimetry for measuring mass changes of mountain glaciers (Foresta et al., 2016; Jakob et al., 2021) and the launch of ICESat-2 –, and new community initiatives, such as GlamBIE (the Glacier mass balance Inter-comparison Exercise), will further contribute to this effort.

**3 iv)** Generating the final reconciled time-series of cumulative mass change: The time series are integrated and cumulative error is root sum square of annual errors, assuming errors are not correlated over time. Errors quoted in the text refer to the one sigma error. I do not understand this, is sigma not the standard deviation? What if errors are correlated over time? What about the large potential systematic error of including peripheral ice masses? These were major concerns by the reviewers.
**Response:** Here sigma refers to the uncertainty estimate and not to the standard deviation. We agree that this is confusing and have removed the sentence 'Errors quoted in the text refer to the one sigma error' and now only show the 1-uncertainty range on Figure 1 and 2 instead of the 1-, 2-, 3-sigma ranges.

We have addressed the systematic error of including the peripheral ice masses in our estimate at the end of Section 3 and have detailed this in our previous comment.

**Actions:**
- We further detailed our error calculation as:

We generate a time-series of cumulative ice sheet mass change by integrating our reconciled time-series of mass trends over time for each region. We estimate the cumulative errors as the root sum square of errors, divided by 12 as our estimates are posted at monthly epochs:

$$\sigma_{cumul}(t) = \sqrt{\frac{1}{12}\sum_{i=0}^{t-1}\sigma_{reconciled}^{2}(i)} \qquad (7)$$

- We modified Figures 1 and 2 to show only the uncertainty range (and removed the 2-, 3-uncertainty ranges).

**P 15, L 304:** Why is Figure 3 discussed before Figure 2? Should Figure 2 not be mentioned and pointed out earlier in the section?
**Action:** We added a mention to Figure 2 earlier in Section 4.

**Figure 3:** What do the bars represent? What is this range? It is not the max and min values as shown in Fig A1, nor does it appear to be the standard deviation. The text says "We report the standard deviation of the aggregated-altimetry, gravimetry and input-output estimates rates of mass change and compare it to the reconciled rate of mass change and its uncertainty (computed as described in Section 3)." Please clarify this.
**Response:** The pink, green, and blue vertical coloured bars represent the rates of mass change over the overlapping period of the three techniques derived from the aggregated time-series of altimetry, gravimetry, and the input-output method, respectively, as calculated from Section 3 step (ii). Their range is their mass balance rates +/- uncertainty. The grey box represents the reconciled estimate calculated from combining these aggregated group estimates following Section 3 step (iii). The horizontal grey bar in the middle of the grey box is the reconciled rate of mass change and the height of the grey box is the uncertainty on the reconciled rate of mass change, calculated from step (iii).

In the text, we contrast the standard deviation in aggregated rates of mass balance over the overlap period of the three techniques (altimetry, gravimetry, input-output over their common periods) to the uncertainty of their reconciled estimate (which is the combination of those three aggregated time-series) over the same periods.

**Actions:**
- We clarified the caption of Figure 3 as:

Inter-comparison of rates of ice sheet mass balance of (a) the AIS, WAIS, EAIS, and APIS over the overlap period 2002-2019 and of (b) the GrIS during the overlap period 2003-2018 derived from the altimetry, gravimetry, and input-output techniques. The coloured bars represent the rates of mass balance and uncertainties of the aggregated technique time-series as calculated in Section 3 step (ii). The grey box represents the rate of mass balance of our final reconciled assessment calculated following the procedure detailed in Section 3 step (iii). The horizontal line in the middle of the box shows the reconciled rate of mass balance and the height of the box represents its associated uncertainty.

- We clarified the corresponding text as:

We compare the standard deviation in aggregated rates of mass change altimetry, gravimetry and input-output estimates rates of mass change and to the uncertainty of our reconciled mass balance estimate (computed from Eq. 5) to assess whether differences between techniques are significant compared to the uncertainty of our reconciled assessment.

**P16, L 320:** It appears the Greenland series show the strongest temporal correlation. There are times when the temporal trends are in opposite direction for WAIS. And is this with reference to Fig A1? Then on line 323 it says the altimetry series is poorly correlated with the other series for GrIS. Judging from Fig A1 this is incorrect. Is this a mistake? Am I missing something?

**Response:** Thank you for spotting this mistake. Yes, the three time-series are well correlated at the GrIS. Here are the correlation coefficients for each ice sheet:

| | $R^2$ (ALT/IOM) | $R^2$ (ALT/GMB) | $R^2$ (GMB/IOM) |
|---|---|---|---|
| **GrIS** | 0.66 | 0.79 | 0.83 |
| **APIS** | 0.11 | 0.18 | 0.83 |
| **WAIS** | 0.36 | 0.52 | 0.83 |
| **EAIS** | 0.02 | 0.05 | 0.32 |

**Action:** We have corrected the text accordingly:

When examining the aggregated time-series of rate of mass change at annual resolution, we find the highest temporal correlation between the three time-series at the GrIS ($0.66 < r^2 < 0.83$). In addition, the gravimetry and input-output annual rates are also well-correlated at the APIS and WAIS ($r^2 = 0.83$). However, the altimetry mass balance time-series is poorly correlated with both the aggregated gravimetry and input-output time-series at the APIS and EAIS ($r^2 < 0.18$).

**P16, L 325-327:** It says that almost all annual mb estimates fall with one standard deviation of the reconciled estimate. Is this by method (altimetry, IO, gravimetry)? Or overall? It seems there is far more variation among individual mass balance estimates than that.

**Response:** We made a mistake in reporting the percentages, we previously reported the proportion of rates falling within 2-uncertainty range. We have corrected this to report the proportion of individual mass changes falling within our uncertainty estimate. We only report the overall proportion of estimates falling within the uncertainty and do not report this per satellite technique as we only have one input-output estimate and thus we feel that reporting the percentages per technique would not be representative.

**Action:** We have corrected the corresponding text as:

Overall, we find that the vast majority of individual estimates of annual rates of mass balance included in this study fall within the uncertainty bounds of our reconciled estimate given their respective individual errors, with 96 %, 83 %, 83 %, 76 %, and 81 % of those annual rates of mass change falling within the reconciled uncertainty range at the GrIS, AIS, APIS, EAIS, and WAIS, respectively.

**Figure 4:** I have been confused by this; is one standard deviation (sigma) the uncertainty or is this the uncertainty computed from step 3 iii)? I do not think that reviewer #2 comment RC2.3 has been adequately addressed and there is still a need to clarify much of these metrics throughout.

**Response:** This is the uncertainty computed from step (iii), we added the formula used to calculate this uncertainty in the revised manuscript. Sigma here refers to the uncertainty and not to the standard deviation. We have clarified this throughout the manuscript and no longer use the notation σ in the different captions and only show the uncertainty estimates instead of different ranges (1*uncertainty, 2*uncertainty, 3*uncertainty). When we make use of the standard deviation it is only

when comparing estimates together to look at the spread in rates of mass balance within or between satellite technique and we never use the standard deviation as an uncertainty estimate.

**Action:** We modified the caption of Figure 4 as:

Cumulative ice sheet mass changes. The shadings represent the associated uncertainties and are calculated following the procedure described in Section 3 (iv). The dashed lines show the results from our previous assessments (IMBIE-2).

**Section 5.2** Comparisons to sea level contribution and projections of future sea level rise: The discussion here is out of scope for this journal and should be cut out along with Figure 5 and Table 3 (see ESSD aims and scope: https://www.earth-system-science-data.net/about/aims_and_scope.html).
**Action:** We have removed this comparison and the related figure and table.

**Section 5.1** is questionable whether it is within scope of ESSD, although it might be beneficial to explain more clearly what was different about the processing scheme in this study—likely in the Methods section—and what has changed in the individual mass balance estimates in Section 2.1. The sea level contributions described in 5.2 can simply be added to Table 2, thus preserving these results and presenting them in a more clear and comprehensive way.
**Response:** We felt the discussion in Section 5.1 was appropriate to include as part of ESSD's living data process – due to the nature of the community submission processes relatively small differences may arise between updated versions of the dataset and thus we believe that it is important to report evolutions of our dataset. As we also foresee future updates to our method (as described in the roadmap) this section will be a natural part of the paper in future, so we include it here to begin the process of tracking the changes that have taken place.

**Action:** In Table 2, we replaced the fractions in GMSL with the sea level contributions.

**Section 5.3** is helpful as it describes limitations of the data. I am not sure that a roadmap for further work is within the scope of ESSD, but I can see the benefit of it here and I'd say it can be kept.
**Response:** Thank you, we have kept this section.

The inclusion of peripheral glaciers is a critical weakness of this analysis and that should be made clear. It would be better to see these limitations presented upfront when describing the methods in Section 3.
**Response:** We have addressed this at the end of Section 3, please see our previous response detailing the changes made.

**P22, L 447:** It says you used 26 mass balance estimates for Greenland and 24 for Antarctica, but this is inconsistent with what is described in Section 2.1 (27 for Greenland, 23 for Antarctica). Please correct this.
**Action:** Thank you, we have corrected this.